# Convergence of Clipped-SGD for Convex $(L_0, L_1)$-Smooth Optimization with Heavy-Tailed Noise

## Abstract

Gradient clipping is a widely used technique in Machine Learning and Deep Learning (DL), known for its effectiveness in mitigating the impact of heavy-tailed noise, which frequently arises in the training of large language models. Additionally, first-order methods with clipping, such as Clip-SGD, exhibit stronger convergence guarantees than SGD under the $(L_0, L_1)$-smoothness assumption, a property observed in many DL tasks. However, the high-probability convergence of Clip-SGD under both assumptions – heavy-tailed noise and $(L_0, L_1)$-smoothness – has not been fully addressed in the literature. In this paper, we bridge this critical gap by establishing the first high-probability convergence bounds for Clip-SGD applied to convex $(L_0, L_1)$-smooth optimization with heavy-tailed noise. Our analysis extends prior results by recovering known bounds for the deterministic case and the stochastic setting with $L_1 = 0$ as special cases. Notably, our rates avoid exponentially large factors and do not rely on restrictive sub-Gaussian noise assumptions, significantly broadening the applicability of gradient clipping.

## 1 Introduction

Stochastic optimization forms the backbone of modern machine learning (Shalev-Shwartz and Ben-David, 2014) and deep learning (Goodfellow et al., 2016), providing the computational efficiency required to train models at scale. While full-gradient methods offer precise optimization, they are often impractical for real-world applications due to their prohibitive computational costs and memory demands. In contrast, Stochastic Gradient Descent (SGD) (Robbins and Monro, 1951) has emerged as the de facto standard for training deep learning models, thanks to its simplicity, scalability, and effectiveness in high-dimensional settings. However, despite its widespread use, SGD alone is often insufficient for capturing the full complexity of modern optimization problems.

Gradient clipping is one of the most widely adopted extensions of SGD, providing a simple yet powerful mechanism for controlling gradient magnitudes in the presence of noisy updates. Clip-SGD and its variants have demonstrated significant practical utility across a range of challenging machine learning tasks. For instance, Pascanu et al. (2013) employed gradient clipping to stabilize the training of recurrent neural networks, which are particularly prone to gradient explosions due to their architectural structure. More recently, gradient clipping has become a crucial component in the training of large language models (LLMs) such as BERT (Devlin et al., 2019), GPT-3 (Brown et al., 2020), Switch Transformers (Fedus et al., 2022), and LLaMA (Touvron et al., 2023).

Gradient clipping is particularly effective in stabilizing the training of deep learning models in the presence of *heavy-tailed noise* in stochastic gradients. This phenomenon, where the probability density of gradient noise decays polynomially, leading to potentially unbounded variance, has been observed in real-world settings such as the pre-training of BERT models (Zhang et al., 2020c). Under such conditions, classical SGD can suffer from divergence, even in expectation, making it poorly suited for training in these high-variance environments. In contrast, gradient clipping not only mitigates these explosive gradient updates but also plays a critical role in establishing *high-probability* convergence guarantees. Recent studies (Gorbunov et al., 2020; Cutkosky and Mehta, 2021; Sadiev et al., 2023; Nguyen et al., 2023; Chezhegov et al., 2024) have shown that employing a clipping threshold that *grows with the number of iterations* can yield high-probability convergence

bounds with only polylogarithmic dependence on the confidence level. In contrast, neither classical SGD nor popular adaptive methods such as AdaGrad (Duchi et al., 2011) and Adam (Kingma and Ba, 2014) can achieve such favorable convergence bounds (Sadiev et al., 2023; Chezhegov et al., 2024), highlighting the advantages of gradient clipping in handling heavy-tailed noise.

Gradient clipping is also particularly well-suited for optimization problems characterized by relaxed smoothness assumptions, which better capture the complex landscapes typical of deep learning. For example, Zhang et al. (2020b) empirically demonstrated that the local smoothness constant along the training trajectory of various deep learning models often scales linearly with the gradient norm. This observation led Zhang et al. (2020b) to the introduction of the more general $(L_0, L_1)$-smoothness assumption, which strictly extends the classical $L$-smoothness by allowing the smoothness constant to depend on the gradient magnitude. This assumption aligns more closely with the real-world behavior of deep learning models, where the loss surface can vary significantly across different regions. Crucially, it has been shown that first-order methods incorporating gradient clipping can achieve faster convergence rates under $(L_0, L_1)$-smoothness compared to their unclipped counterparts (Zhang et al., 2020b;a; Koloskova et al., 2023; Gorbunov et al., 2025; Vankov et al., 2025). However, a critical aspect of all these results is the careful selection of the clipping threshold, which must be set as a specific *constant determined by the parameters $L_0$ and $L_1$*.

This observation highlights a fundamental mismatch in the design of gradient clipping strategies: under the $(L_0, L_1)$-smoothness assumption, the clipping threshold is typically set as a fixed constant determined by problem-specific parameters ($L_0$ and $L_1$), while in the presence of heavy-tailed noise, the threshold is often required to grow with the total number of iterations to ensure stability and convergence. This apparent conflict raises a critical open question:

> *How should the clipping threshold be chosen to effectively address*
>
> *heavy-tailed noise and $(L_0, L_1)$-smoothness?*

**Our contribution.** In this paper, we resolve the above open question by providing the first high-probability convergence analysis of Clip-SGD under the joint assumptions of heavy-tailed noise and $(L_0, L_1)$-smoothness. Specifically, for convex $(L_0, L_1)$-smooth problems with stochastic gradients having bounded central $\alpha$-th moment for some $\alpha \in (1, 2]$, we establish a high-probability convergence rate of

$$\tilde{\mathcal{O}} \left( \max \left\{ \frac{L_0 R_0^2}{K}, \frac{\max\{1, L_1 R_0\} R_0 \sigma}{K^{(\alpha-1)/\alpha}} \right\} \right),$$

where $\tilde{\mathcal{O}}$ hides numerical and polylogarithmic factors, $K = \tilde{\Omega} \left( \frac{(L_1 R_0)^{2+\alpha}}{\delta} \right)$ is the number of iterations required to achieve a high-probability bound with confidence level $1 - \delta$, and $R_0$ is an upper bound on the initial distance to the solution. Our result not only recovers the known deterministic convergence rates for generalized smoothness (Gorbunov et al., 2025; Vankov et al., 2025), but also fully reproduces the stochastic convergence guarantees in the special case of $L_1 = 0$ (Sadiev et al., 2023). Importantly, our analysis avoids the exponentially large factors that can arise from the generalized smoothness assumption, marking a significant improvement over previous approaches. For a detailed comparison, see Table 1.

## 2 PRELIMINARIES

**Notation.** The Euclidean norm in $\mathbb{R}^d$ is denoted as $\|x\| = \sqrt{\langle x, x \rangle}$. The norm $\|X\|_2$, where $X \in \mathbb{R}^{d \times d}$, is the spectral norm of the matrix. The $\mathbb{E}_\xi[\cdot]$ denotes the expectation w.r.t. random variable $\xi$. The ball with the center at $x \in \mathbb{R}^d$ and radius $r$ is defined as a $B_r(x) := \{y \in \mathbb{R}^d | \|x - y\| \le r\}$. The clipping operator is denoted as $\mathtt{clip}(x, \lambda) := \min \left\{ 1, \frac{\lambda}{\|x\|} \right\} x$. We often use $R_0$ to denote some upper bound on the distance between the starting point and the solution of the problem.

**Problem.** We focus on the classical stochastic optimization problem, which can be stated as

$$\min_{x \in \mathbb{R}^d} \{f(x) := \mathbb{E}_{\xi \sim \mathcal{D}}[f(x, \xi)]\}. \tag{1}$$

This formulation is foundational in machine learning (Shalev-Shwartz and Ben-David, 2014), where $f$ represents the loss function of the model, $x$ are the parameters to be optimized, $\mathcal{D}$ is the underlying

Table 1: Comparison of the state-of-the-art high-probability convergence results for Clip-SGD applied to convex problems satisfying the heavy-tailed noise assumption (Assumption 3) and/or $(L_0, L_1)$-smoothness assumption (Assumption 2).

| Reference | $L$-smooth | $(L_0, L_1)$-smooth | Stochasticity | | Clipping level | Complexity |
|---|---|---|---|---|---|---|
| | | | Light tails | Heavy tails | | |
| Sadiev et al. (2023) | ✓ | ✓✗[1] | ✓ | ✓ | $\Theta\left(\max\{LR_0, \sigma K^{1/\alpha}\}\right)$ | $\tilde{\mathcal{O}}\left(\frac{LR_0^2}{K} + \frac{R_0\sigma}{K^{(\alpha-1)/\alpha}}\right)$ |
| Gorbunov et al. (2025) Vankov et al. (2025) Lobanov et al. (2024) | ✓ | ✓ | ✗[2] | ✗ | $\Theta\left(\frac{L_0}{L_1}\right)$ | $\mathcal{O}\left(\frac{LR_0^2}{K}\right)^{[3]}$ |
| Gaash et al. (2025) | ✓ | ✓ | ✓ | ✗[4] | $\Theta\left(\max\left\{\frac{L_0}{L_1}, \frac{\sigma\sqrt{K}}{L_1 R_0}\right\}\right)$ | $\tilde{\mathcal{O}}\left(\frac{L_0 R_0^2}{K} + \frac{R_0\sigma}{\sqrt{K}} + (L_1 R_0)^2\right)$ |
| **This work** | ✓ | ✓ | ✓ | ✓ | $\Theta\left(\max\left\{\frac{L_0}{L_1}, \sigma K^{1/\alpha}\right\}\right)$ | $\tilde{\mathcal{O}}\left(\max\left\{\frac{L_0 R_0^2}{K}, \frac{\max\{1, L_1 R_0\}R_0\sigma}{K^{(\alpha-1)/\alpha}}\right\}\right)^{[5]}$ |

[1] Sadiev et al. (2023) make all assumptions on a ball centered at $x^*$ and having radius $\sim 2R_0$ and show that the iterates do not escape this ball with high probability. On such a set, $(L_0, L_1)$-smoothness implies $L$-smoothness with $L = L_0(1 + L_1 R_0 \exp(L_1 R_0))$, making the final bound dependent on the exponentially large factor of $L_1 R_0$.
[2] Deterministic result.
[3] Gorbunov et al. (2025) prove this bound for $K = \Omega((L_1 R_0)^2)$, while Vankov et al. (2025); Lobanov et al. (2024) obtain it for $K = \tilde{\Omega}(L_1 R_0)$.
[4] Gaash et al. (2025) derive their result under the assumption that the noise is sub-Gaussian (3).
[5] This bound holds for $K = \tilde{\Omega}\left(\frac{(L_1 R_0)^{2+\alpha}}{\delta}\right)$.

data distribution, and $\xi$ captures the stochasticity introduced by sampling the data. We consider Clip-SGD (Algorithm 1) applied to this problem.

---

**Algorithm 1** Clip-SGD

1: **Input:** Starting point $x_0$, level of clipping $\lambda$, learning rate $\gamma$
2: **for** $k = 0, \ldots, K - 1$ **do**
3:     Sample $\nabla f(x_k, \xi_k)$
4:     $x_{k+1} = x_k - \gamma \texttt{clip}(\nabla f(x_k, \xi_k), \lambda)$
5: **end for**

---

**Assumptions.** In this part, we introduce and briefly discuss the assumptions used in the analysis. First, let us introduce the assumption of convexity.

**Assumption 1** (Convexity). *The function $f$ is convex, i.e., for all $x, y \in \mathbb{R}^d$ the next inequality holds:*

$$f(y) \geq f(x) + \langle \nabla f(x), y - x \rangle.$$

Next, we will use the assumption of $(L_0, L_1)$-smoothness.

**Assumption 2** ($(L_0, L_1)$-smoothness). *The function $f$ is $(L_0, L_1)$-smooth, i.e. for all $x, y \in \mathbb{R}^d$ the next inequality holds:*

$$\| \nabla f(x) - \nabla f(y) \| \leq \left( L_0 + L_1 \sup_{u \in [x,y]} \|\nabla f(u)\| \right) \| x - y \|.$$

Historically, the first version of the above assumption was formulated by Zhang et al. (2020b) for twice differentiable functions as follows:

$$\left\| \nabla^2 f(x) \right\|_2 \leq L_0 + L_1 \|\nabla f(x)\|, \quad x \in \mathbb{R}^d.$$

Later, it was generalized to the case of functions not necessarily having second derivatives by Zhang et al. (2020a). Assumption 2 was first introduced by Chen et al. (2023), and it is equivalent to the one proposed by Zhang et al. (2020a). This assumption is strictly more general than

$$\|\nabla f(y) - \nabla f(x)\| \leq L \|y - x\|, \quad \forall x, y \in \mathbb{R}^d, \tag{2}$$

known as $L$-smoothness: it reduces to the standard $L$-smoothness with $L = L_0$ if $L_1 = 0$. Moreover, one can construct functions that satisfy Assumption 2 but not $L$-smoothness, e.g., exponent of norm $f(x) = \exp(\|x\|) + \exp(-\|x\|)$, power of norm $f(x) = \|x\|^n$ for $n > 2$, and exponent of the linear function $f(x) = \exp(\langle a, x \rangle)$ (Chen et al., 2023; Gorbunov et al., 2025).

Finally, we assume unbiasedness and boundedness of the $\alpha$-*th* central moment.

**Assumption 3** (Stochastic oracle). *The stochastic oracle $\nabla f(x, \xi)$ is unbiased and have bounded $\alpha$-th central moment with $\alpha \in (1, 2]$, i.e.*

$$\mathbb{E}[\nabla f(x, \xi)] = \nabla f(x); \qquad \mathbb{E}[\|\nabla f(x, \xi) - \nabla f(x)\|^\alpha] \leq \sigma^\alpha.$$

This assumption has become relatively standard – it has already been considered in (Zhang et al., 2020c; Cutkosky and Mehta, 2021; Sadiev et al., 2023; Nguyen et al., 2023; Chezhegov et al., 2024). Prominent examples of distributions that satisfy Assumption 3 include Lévy $\alpha$-stable noise, as well as synthetic one-dimensional distributions that can be easily constructed. In turn, case $\alpha = 2$ corresponds to one of the most classical assumptions on the stochastic oracle (Nemirovski et al., 2009; Ghadimi and Lan, 2013; Takáč et al., 2013).

**High-probability convergence bounds.** A vast body of work in stochastic optimization has focused on establishing convergence guarantees in expectation. Specifically, for an iterative process $\{x_k\}_{k=0}$ and a target criterion $C(\{x_k\})$, the typical goal is to identify the smallest number of iterations $K$ needed to ensure that $\mathbb{E}\left[C\left(\{x_k\}_{k=0}^{K-1}\right)\right] \leq \varepsilon$ is satisfied. However, this expectation-based approach only captures the average performance of the algorithm and does not fully reflect the variability inherent in the stochastic process. In contrast, high-probability bounds, which ensure that the desired criterion is satisfied with high confidence, are often more informative. These bounds take the form $\mathbb{P}\left\{C\left(\{x_k\}_{k=0}^{K-1}\right) \leq \varepsilon\right\} \geq 1 - \delta$, directly controlling the likelihood of worst-case deviations.

While it is possible to derive high-probability bounds from expectation bounds using tools like Markov's inequality, this approach typically results in convergence rates with an *inverse-power* dependence on $\delta$. Modern methods aim for much tighter, *polylogarithmic* dependence on $\frac{1}{\delta}$, which significantly reduces the required number of iterations for a given confidence level. Achieving this improved scaling generally requires either imposing stronger assumptions, e.g., sub-Gaussian noise

$$\mathbb{E}\left[\exp\left(\|\nabla f(x,\xi) - \nabla f(x)\|^2 / \sigma^2\right)\right] \leq \exp(1), \tag{3}$$

or employing advanced techniques such as gradient clipping, truncation, or normalization.

## 3 RELATED WORK

**Convergence under $(L_0, L_1)$-smoothness.** Early studies on the convergence of first-order methods under $(L_0, L_1)$-smoothness has primarily focused on the non-convex setting (Zhang et al., 2020b;a; Zhao et al., 2021; Chen et al., 2023; Hübler et al., 2024b; Khirirat et al., 2024; Crawshaw et al., 2022; Faw et al., 2023; Wang et al., 2022; 2023; Li et al., 2024; Bilel, 2024; Liu and Zhou, 2024; Vankov et al., 2025), which we discuss in Appendix A. In the convex setting, the analysis is more recent and less developed. Koloskova et al. (2023) provided convergence guarantees for Clip-GD under convexity, $(L_0, L_1)$-smoothness and $L$-smoothness, deriving a complexity bound of $\mathcal{O}\left(\max\left\{(L_0 + \lambda L_1)R_0^2/\varepsilon, \sqrt{R_0^4 L(L_0 + \lambda L_1)^2/\lambda^2 \varepsilon}\right\}\right)$. The leading term in this complexity bound is independent of $L_1$ and $L$, if $\lambda \sim L_0/L_1$, and can significantly outperform standard GD. Building on this, Takezawa et al. (2024) analyzed GD with Polyak stepsizes and derived $\mathcal{O}\left(\max\left\{L_0 R_0^2/\varepsilon, \sqrt{R_0^4 LL_1/\varepsilon}\right\}\right)$ complexity bound. Li et al. (2023) considered GD and Nesterov's accelerated gradient (Nesterov, 1983) under the broad class of functions satisfying the so-called $(r, \ell)$-smoothness and derive $\mathcal{O}\left(\ell R_0^2/\varepsilon\right)$ and $\mathcal{O}\left(\sqrt{\ell R_0^2/\varepsilon}\right)$ complexities respectively, where $\ell := L_0 + L_1 G$ and $G$ is dependent on smoothness parameters $(L_0, L_1)$, initial gradient norm, and functional suboptimality. However, through the constants $L$ and $G$, the bounds from Koloskova et al. (2023); Takezawa et al. (2024); Li et al. (2023) include exponentially large factors of $L_1 R_0$, a significant drawback addressed by the more recent results of Gorbunov et al. (2025); Vankov et al. (2025); Lobanov et al. (2024), which currently provide the tightest known bounds for deterministic convex $(L_0, L_1)$-smooth problems. Additionally, Tyurin (2024) present a unified analysis of GD (with specific stepsizes) for both convex and non-convex problems under a more general $\ell(\|\nabla f(x)\|)$-smoothness condition, and Yu et al. (2025b) study Mirror Descent and its variants under a version of $(r, \ell)$-smoothness (Li et al., 2023), adapted to non-Euclidean norms.

Most of the works discussed above also present the convergence results for the stochastic methods (Zhang et al., 2020b;c; Zhao et al., 2021; Chen et al., 2023; Crawshaw et al., 2022; Faw et al., 2023; Wang et al., 2022; 2023; Li et al., 2024; Hübler et al., 2024b; Gorbunov et al., 2025; Yu et al., 2025b). In addition, Yang et al. (2024) propose and analyze a variant of Normalized SGD with independent normalization. Yu et al. (2025a) establish new convergence results for an accelerated version of SGD with both constant and adaptive stepsizes under $(L_0, L_1)$-smoothness and relaxed affine variance assumptions. Furthermore, Tovmasyan et al. (2025) introduce a generalized smoothness condition

called $\psi$-smoothness and derive new convergence bounds for the Stochastic Proximal Point Method (Bertsekas, 2011) under this framework. However, these papers do not address the heavy-tailed noise settings, and only Faw et al. (2023); Wang et al. (2023); Li et al. (2024); Yu et al. (2025a) provide high-probability convergence guarantees. However, the bounds from Faw et al. (2023); Wang et al. (2023) have inverse-power dependencies on $\delta$, while the results of Li et al. (2024); Yu et al. (2025a) rely on a sub-Gaussian noise assumption (3)[1].

**High-probability convergence under the light-tailed noise.** High-probability convergence guarantees have long been a critical component in the analysis of stochastic first-order methods, particularly when the noise in the stochastic gradients is light-tailed. In these settings, methods like SGD and its variants can achieve convergence rates with the polylogarithmic dependence on the failure probability $\delta$. Under the sub-Gaussian noise assumption, this behavior has been rigorously established for SGD (Nemirovski et al., 2009; Harvey et al., 2019), its accelerated counterparts (Ghadimi and Lan, 2012; Dvurechensky and Gasnikov, 2016), and adaptive methods like AdaGrad (Li and Orabona, 2020; Liu et al., 2023). Recent extensions to even broader classes of noise, such as sub-Weibull distributions, have further expanded this theoretical framework (Madden et al., 2024).

The most closely related work to ours is that of Gaash et al. (2025), who derive high-probability convergence rates with polylogarithmic dependence on $\delta$ for convex $(L_0, L_1)$-smooth optimization under the assumption of sub-Gaussian noise in the stochastic gradients. Their approach involves a variant of Clip-SGD that uses two independent stochastic gradients – one for the update direction and another for the clipping multiplier. While this technique effectively avoids the exponentially large factors of $L_1 R_0$, its performance in the presence of heavy-tailed noise remains unclear.

**High-probability convergence under the heavy-tailed noise.** Gradient clipping is one of the most popular approaches to deal with the heavy-tailed noise in the literature on the high-probability convergence. Early work in this direction includes the truncated Stochastic Mirror Descent method proposed by Nazin et al. (2019), which established high-probability complexity bounds for convex and strongly convex problems under the bounded variance assumption (Assumption 3 with $\alpha = 2$). Building on this foundation, Gorbunov et al. (2020) provided the first comprehensive high-probability bounds for Clip-SGD (Algorithm 1) and introduced an accelerated variant using the Stochastic Similar Triangles Method (SSTM) (Gasnikov and Nesterov, 2016). Subsequent work extended these results to broader problem classes, including non-smooth optimization (Gorbunov et al., 2024a; Parletta et al., 2024), unconstrained variational inequalities (Gorbunov et al., 2022), and problems satisfying Assumption 3 with $\alpha < 2$ (Cutkosky and Mehta, 2021; Sadiev et al., 2023; Nguyen et al., 2023; Gorbunov et al., 2024b). Adaptive variants have also been developed: Li and Liu (2023) analyzed Clip-AdaGrad with scalar stepsizes, while Chezhegov et al. (2024) obtained similar bounds for both scalar and coordinate-wise versions of Clip-AdaGrad and Clip-Adam. In the zeroth-order setting, Kornilov et al. (2023) proposed a clipped variant of SSTM. For distributed setup, Lee et al. (2025) provided in-expectation convergence rates for the TailOPT method. Moreover, standard clipping was successfully adapted to address differential privacy (Khah et al., 2025) with high-probability convergence.

Beyond gradient clipping, several alternative strategies for achieving high-probability convergence have been proposed. These include robust distance estimation with inexact proximal point methods (Davis et al., 2021), gradient normalization (Cutkosky and Mehta, 2021; Hübler et al., 2024a), and sign-based methods (Kornilov et al., 2025). Notably, some of these approaches, such as those proposed by Hübler et al. (2024a) and Kornilov et al. (2025), do not require prior knowledge of the tail parameter $\alpha$, albeit at the cost of sub-optimal convergence rates. For symmetric distributions, recent work has provided high-probability guarantees for non-linear transformations like standard clipping, coordinate-wise clipping, and normalization (Armacki et al., 2023; 2024), while Puchkin et al. (2024) has explored median-based clipping under structured non-symmetric noise.

Despite these advancements, existing high-probability convergence results for the $(L_0, L_1)$-smooth case (with the heavy-tailed noise) still suffer from the presence of exponentially large factors involving $L_1 R_0$ in their bounds.

---

[1]Yu et al. (2025a) use a more general version of (3) with $\sigma^2 = A(f(x) - f(x^*)) + B \|\nabla f(x)\| + C$.

## 4 MAIN RESULT

In this section, we provide our main convergence result for Clip-SGD method (Algorithm 1). The next theorem provides new high-probability convergence rates for Clip-SGD.

**Theorem 1.** *Suppose that Assumptions 1, 2 and 3 hold. Then, after $K$ iterations of Clip-SGD (Algorithm 1) with*

$$\lambda = \max\left\{2L_0 \min\left\{4R_0, \frac{1}{L_1}\right\}, 9^{\frac{1}{\alpha}}\sigma K^{\frac{1}{\alpha}}\left(\ln\left(\frac{4K}{\delta}\right)\right)^{-\frac{1}{\alpha}}\right\},$$

$$\gamma = \frac{1}{160\lambda \ln\left(\frac{4K}{\delta}\right)} \min\left\{4R_0, \frac{1}{L_1}\right\},$$

*we have:*

- *If $4R_0 \leq \frac{1}{L_1}$, then*

$$f\left(\frac{1}{K}\sum_{k=0}^{K-1} x_k\right) - f^* = \tilde{\mathcal{O}}\left(\max\left\{\frac{L_0 R_0^2}{K}, \frac{R_0\sigma}{K^{(\alpha-1)/\alpha}}\right\}\right)$$

  *with probability at least $1 - \delta$.*

- *If $4R_0 \geq \frac{1}{L_1}$ and $K = \Omega\left(\frac{(L_1 R_0)^{2+\alpha}\ln^2\left(\frac{K}{\delta}\right)}{\delta}\right)$*

$$\min_{k=0,\ldots,K-1}(f(x_k) - f^*) = \tilde{\mathcal{O}}\left(\max\left\{\frac{L_0 R_0^2}{K}, \frac{L_1 R_0^2 \sigma}{K^{(\alpha-1)/\alpha}}\right\}\right)$$

  *holds with probability at least $1 - \delta$.*

*Proof sketch.* The proof begins with the establishment of a descent lemma (Lemma 3, Appendix B), formulated in a case-based manner to account for the various possible relationships between $\|\nabla f(x_k)\|$, the clipping threshold $\lambda$, and the ratio $\frac{L_0}{L_1}$, in line with existing analyses under $(L_0, L_1)$-smoothness (Koloskova et al., 2023; Takezawa et al., 2024; Gorbunov et al., 2025). Following the approach of Sadiev et al. (2023), we define a sequence of events $E_k$, which imply the main result for $k = K$. We then use an inductive argument to derive sufficiently strong lower bounds on the probabilities of these events, proving by induction that $\mathbb{P}\{E_k\} \geq 1 - \frac{k\delta}{K}$, which yields the desired bound for $k = K$. However, our proof introduces an additional layer of complexity by distinguishing two separate cases based on the relationship between the initial distance to the optimum, $R_0$, and $\frac{1}{L_1}$.

In the first case $\left(4R_0 \leq \frac{1}{L_1}\right)$, our proof follows from the result from Sadiev et al. (2023) (though we provide the full proof for the convenience). This is expected, as we show that with high probability, the iterates remain within the ball $B_{\sqrt{2}R_0}(x^*)$. Consequently, for any $x, y$ within this set, the terms $L_1\|\nabla f(x)\|$ and $\exp(L_1\|y - x\|)$ from Proposition 2 can be bounded by $\mathcal{O}(L_0)$ and $\mathcal{O}(1)$, respectively, implying that the objective function is $L$-smooth on $B_{\sqrt{2}R_0}(x^*)$ with $L = \mathcal{O}(L_0)$.

In contrast, in the second case ($4R_0 \geq \frac{1}{L_1}$), we must additionally control the effect of rare, large gradient norms that exceed the clipping threshold. Specifically, for any $0 < T \leq K$, we show that $E_{T-1}$ implies

$$\sum_{l \in T_1(t) \cup T_2(t)} \gamma(f(x_l) - f^*) \leq \|x_0 - x^*\|^2 - \|x_t - x^*\|^2 \tag{4}$$

$$- \sum_{l \in T_1(t) \cup T_2(t)} 2\gamma\langle\theta_l, x_l - x^*\rangle + \sum_{l \in T_1(t) \cup T_2(t)} 2\gamma^2\|\theta_l\|^2 \tag{5}$$

$$- \sum_{l \in T_3(t)} 2\gamma\langle\hat{\theta}_l, x_l - x^*\rangle - \frac{\gamma\lambda|T_3(t)|}{16L_1} \tag{6}$$

holds for $t = 1, \ldots, T$, where $\theta_l := \texttt{clip}(\nabla f(x_l, \xi_l), \lambda) - \nabla f(x_l)$, $\hat{\theta}_l := \texttt{clip}(\nabla f(x_l, \xi_l), \lambda) - \texttt{clip}(\nabla f(x_l), \lambda/2)$, and

$$T_1(t) := \left\{ k \in 0, \ldots, t-1 \mid \|\nabla f(x_k)\| \le \frac{L_0}{L_1} \right\},$$

$$T_2(t) := \left\{ k \in 0, \ldots, t-1 \mid \frac{\lambda}{2} \ge \|\nabla f(x_k)\| > \frac{L_0}{L_1} \right\},$$

$$T_3(t) := \left\{ k \in 0, \ldots, t-1 \mid \|\nabla f(x_k)\| > \frac{\lambda}{2} \right\}.$$

As in the first case, we bound the contributions from (4) and (5) by $\mathcal{O}(R_0)$ with high probability using Bernstein's inequality, along with assumptions on $\gamma$ and $\lambda$. However, the key term in (6) is bounded using a different argument. Specifically, we show that the inequality $-2\gamma\langle \hat{\theta}_l, x_l - x^* \rangle \le \frac{\gamma\lambda}{32L_1}$ follows from the condition $\|\xi_l\| \le B := \frac{\lambda}{128L_1R_0}$ for $l \in T_3(t)$, where we slightly abuse notation by defining $\xi_l := \nabla f(x_l, \xi_l) - \nabla f(x_l)$. Furthermore, the construction of $E_{T-1}$ guarantees that $|T_3(T-1)| \le C_1 := 10240(L_1R_0)^2 \ln\left(\frac{4K}{\delta}\right)$, since $E_{T-1}$ also implies $0 \le 2R_0^2 - \frac{\gamma\lambda|T_3(T-1)|}{32L_1}$. To complete the inductive step, we apply Markov's inequality to estimate $\mathbb{P}\{\|\xi_{k-1}\| \le B\}$ under the conditions $k - 1 \in T_3(k)$ and $|T_3(k-1)| \le C_1 - 1$. This step leads to the requirement $K = \Omega\left(\frac{(L_1R_0)^{2+\alpha}\ln^2\left(\frac{K}{\delta}\right)}{\delta}\right)$, which arises from applying Markov's inequality up to $C_1$ times.

Finally, we emphasize that in the second case ($4R_0 \ge \frac{1}{L_1}$), we prove by induction that

$$\mathbb{P}\{E_k\} \ge 1 - \frac{k\delta}{K} - \sum_{r=0}^{k} \min\left\{ \frac{r}{C_1}, 1 \right\} \delta\mathbb{P}\{|T_3(k)| = r\},$$

which significantly differs from the induction assumptions used in previous works (Gorbunov et al., 2020; Sadiev et al., 2023; Gorbunov et al., 2024b). For complete technical details, we refer the reader to Appendix B. $\qquad\square$

## 5 DISCUSSION OF THE RESULT

In this section, we discuss our main convergence results, highlighting their significance in the context of existing work, including a detailed comparison with prior analyses, and addressing the challenges associated with heavy-tailed noise and generalized smoothness.

### 5.1 COMPARISON WITH GAASH ET AL. (2025)

The closest related work to ours is the recent study by Gaash et al. (2025), which also analyzes the high-probability convergence of Clip-SGD under generalized smoothness conditions. Prior to conducting the comparison, we introduce the algorithm (see Algorithm 2) under consideration in (Gaash et al., 2025). For simplicity, we omit the projection operator on some set $\mathcal{X}$ from the original version since it is unnecessary for the convergence guarantees of Algorithm 2.

---

**Algorithm 2** Clip-SGD with double sampling (Gaash et al., 2025)

1: **Input:** Start point $x_0$, level of clipping $\lambda$, learning rate $\gamma$
2: **for** $k = 0, \ldots, K-1$ **do**
3:     Sample $\nabla f(x_k, \xi_k^c), \nabla f(x_k, \xi_k)$ independently
4:     $x_{k+1} = x_k - \gamma \min\left\{ 1, \frac{\lambda}{\|\nabla f(x_k, \xi_k^c)\|} \right\} \nabla f(x_k, \xi_k)$
5: **end for**

---

**Light-tailed noise.** The analysis from Gaash et al. (2025) is restricted to the case of sub-Gaussian noise, which is substantially lighter-tailed than the noise distributions considered in our work (Zhang

et al., 2020c). This assumption simplifies the convergence analysis, as sub-Gaussian noise is inherently more amenable to standard concentration inequalities. In contrast, we focus on the more challenging setting of heavy-tailed noise, characterized by only a bounded central $\alpha$-th moment, which introduces significant technical difficulties in establishing high-probability guarantees.

**Role of clipping.** Furthermore, under the simpler $L$-smoothness assumption (2), the need for clipping in the light-tailed noise setting largely disappears. In this case, the inherent concentration of sub-Gaussian noise is often sufficient to control the gradient norms, making clipping unnecessary. However, when the generalized $(L_0, L_1)$-smoothness assumption is introduced, clipping becomes essential even with light-tailed noise, as it restricts the range of gradient norms, ensuring the validity of the generalized smoothness assumption. In contrast, for heavy-tailed noise, the clipping threshold $\lambda$ must address two competing objectives: (i) it must remain constant to effectively control the gradient norms for the application of the $(L_0, L_1)$-smoothness condition, and (ii) it must scale with the number of iterations to mitigate the impact of rare, extreme gradients. Our analysis demonstrates that standard clipping can simultaneously address both of these challenges, a property that is unnecessary in the purely light-tailed regime where gradient norms are naturally more controlled.

**Practicality.** Finally, the algorithm analyzed in Gaash et al. (2025) employs a double-sampling strategy, where the gradient direction and the clipping threshold are computed using two independent samples. This approach, while providing strong theoretical guarantees, can significantly increase the computational cost and memory requirements, potentially limiting its practical applicability in large-scale machine learning problems. In contrast, our analysis considers the standard, single-sample variant of Clip-SGD, demonstrating that strong convergence guarantees can be obtained without requiring such algorithmic modifications. This distinction is critical, as it reflects a more realistic scenario for practical applications, where computational efficiency is a key concern.

**Upper bounds.** Our main result establishes the following upper bound on the convergence rate:

$$\tilde{\mathcal{O}}\left(\max\left\{\frac{L_0 R_0^2}{K}, \frac{\max\{1, L_1 R_0\} R_0 \sigma}{K^{(\alpha-1)/\alpha}}\right\}\right) \text{ with } K = \Omega\left(\frac{(L_1 R_0)^{2+\alpha} \ln^2\left(\frac{K}{\delta}\right)}{\delta}\right).$$

This result recovers several known special cases from the literature. When $L_1 = 0$, the bound simplifies to the convergence rate for $L$-smooth settings previously established in Sadiev et al. (2023), which corresponds to the classical smooth optimization framework. On the other hand, if the noise level is zero (i.e., $\sigma = 0$), our bound reduces to the deterministic convergence rates derived in the context of GD with smoothed gradient clipping by Gorbunov et al. (2025).

For comparison, the recent work by Gaash et al. (2025) obtained an upper bound of the form

$$\tilde{\mathcal{O}}\left(\max\left\{\frac{L_0 R_0^2}{K}, \frac{R_0 \sigma}{\sqrt{K}}\right\}\right) \text{ with } K = \Omega\left(\ln\left(\frac{K}{\delta}\right)(L_1 R_0)^2\right).$$

While this bound shares a similar structure to ours, their lower bound on $K$ is $(L_1 R_0)^\alpha$ times smaller. This difference arises from the different ways in which gradient clipping manages extreme gradient magnitudes, as discussed in the paragraph on the role of clipping from the previous subsection. Furthermore, the lower bound on $K$ in (Gaash et al., 2025) does not explicitly include a $1/\delta$ factor, due to their reliance on sub-Gaussian noise assumptions, which provide inherently stronger tail control (see equation (3)). In contrast, our analysis, which handles the more general heavy-tailed noise case, requires the use of Markov's inequality to control the probability of rare, high-magnitude gradient events (when $\|\nabla f(x_k)\| \geq \frac{\lambda}{2} \geq \frac{L_0}{L_1}$), leading to a stricter dependence on $\delta$.

Nevertheless, the term proportional to $1/\delta$ in our result has only a polylogarithmic dependence on $K$. This means that our result ensures that $\min_{k=0,\ldots,K-1}(f(x_k) - f^*) \leq \varepsilon$ holds with probability at least $1 - \delta$ after

$$K = \tilde{\mathcal{O}}\left(\max\left\{\frac{L_0 R_0^2}{\varepsilon}, \left(\frac{\max\{1, L_1 R_0\} R_0 \sigma}{\varepsilon}\right)^{\frac{\alpha}{\alpha-1}}, \frac{(L_1 R_0)^{2+\alpha}}{\delta}\right\}\right) \text{ iterations.}$$

### 5.2 On the In-Expectation Bounds and the Choice of the Iterate

**Is our bound stronger than the in-expectation result?** It is natural that the bound we provide may initially appear counterintuitive – the factor $1/\delta$ is indeed non-standard for the high-probability

convergence results. However, at present, no in-expectation results that we could compare to are available for the class of problems under consideration.

Let us, however, assume the existence of an in-expectation bound of the form $\mathbb{E}\left[f(x_K) - f^*\right] \leq \varepsilon$ with $K = K(\varepsilon) = \mathcal{O}\left(\frac{1}{\varepsilon} + \frac{1}{\varepsilon^{\frac{\alpha}{\alpha-1}}}\right)$ where we ignore the problem parameters $L_0, L_1, R_0, \sigma$. This rate in fact coincides with the best-known results for the convex $L$-smooth case (Sadiev et al., 2023; Nguyen et al., 2023). To obtain a high-probability bound from this, the only general tool is Markov's inequality, which would yield $f(x_K) - f^* \leq \varepsilon$ with probability at least $1 - \delta$, provided that $K = K(\varepsilon\delta) = \mathcal{O}\left(\frac{1}{\varepsilon\delta} + \frac{1}{(\varepsilon\delta)^{\frac{\alpha}{\alpha-1}}}\right)$. In contrast, our result guarantees $K = \tilde{\mathcal{O}}\left(\frac{1}{\varepsilon} + \frac{1}{\varepsilon^{\frac{\alpha}{\alpha-1}}} + \frac{1}{\delta}\right)$, which is *strictly better* for all $\delta \in (0, 1)$.

More precisely, the inverse-power dependence on $\delta$ appears only in the term that is independent of $\varepsilon$ (up to logarithmic factors). This means that, unless $\delta$ is much smaller than $\varepsilon$, the $\delta$-dependent term is not dominant – the second term is. Consequently, if an in-expectation bound for the convex $(L_0, L_1)$-smooth case exists, which would almost certainly yield $K = \mathcal{O}\left(\frac{1}{\varepsilon} + \frac{1}{\varepsilon^{\frac{\alpha}{\alpha-1}}}\right)$ up to a problem-dependent constants, then our bound is *strictly stronger* than one obtained via such an expectation-based result.

**How to choose the final iterate?** From a practical perspective, one may ask: which point should be chosen as the final output? The convergence guarantees allow us to take either $\min_{k=0,\ldots,K-1}(f(x_k) - f^*) \leq \varepsilon$ (see Theorem 1), or, as follows from the proof, a point selected through ergodicity over $T_1(K) \cup T_2(K)$. However, two issues arise in practice: i) computing the minimum requires evaluating the full model at every step, which is prohibitively expensive; ii) the sets $T_1(K)$ and $T_2(K)$ are stochastic, and their exact time indices are unknown.

In machine learning and deep learning applications, however, it is crucial to produce concrete model weights that can be directly deployed. To address this, we propose a robust method for selecting the final iterate without weakening the convergence guarantees. Our approach is based on uniform sampling combined with robust estimation of function values. Details are provided in Appendix C.

## 6 CONCLUSION

In this paper, we presented the first high-probability convergence analysis for Clip-SGD under the joint assumptions of heavy-tailed noise and $(L_0, L_1)$-smoothness. Our results establish that for convex $(L_0, L_1)$-smooth optimization problems with stochastic gradients having bounded central $\alpha$-th moment with $\alpha \in (1, 2]$, Clip-SGD with specifically selected clipping level achieves a high-probability convergence rate of

$$\tilde{\mathcal{O}}\left(\max\left\{\frac{L_0 R_0^2}{K}, \frac{\max\{1, L_1 R_0\}R_0\sigma}{K^{(\alpha-1)/\alpha}}\right\}\right) \text{ for } K = \tilde{\Omega}\left(\frac{(L_1 R_0)^{2+\alpha}}{\delta}\right).$$

Our approach successfully avoids the exponentially large factors of $L_1 R_0$.

While our work resolves a critical gap in the convergence theory of stochastic gradient methods under generalized smoothness and heavy-tailed noise, several important open questions remain. First, it would be interesting to investigate the optimality of the lower bound on $K$, i.e., its dependence on $\delta$. Second, it would be valuable to extend these high-probability convergence results to the accelerated methods, such as the ones based on Nesterov's momentum, which are known to exhibit faster convergence under classical smoothness. Third, our analysis is limited to convex optimization, and extending these results to the non-convex case remains a significant challenge, especially under heavy-tailed noise. Fourth, understanding how these techniques can be adapted to handle more complex structures, such as variational inequalities and saddle-point problems, represents another promising direction for future research. Finally, the application of these methods in distributed and federated learning, where the gradient noise can vary significantly across nodes, is another important open problem, particularly in light of recent interest in scalable, decentralized optimization methods.

We hope that our results inspire further research in these directions and contribute to the broader understanding of stochastic optimization under realistic noise and smoothness assumptions.

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

# A    NOTATION TABLE, AUXILIARY FACTS, AND EXTRA RELATED WORK

Table 2: Auxiliary notation used in the proofs.

| Symbol | Formula |
|--------|---------|
| $g_t$ | $\min\left\{1, \frac{\lambda}{\|\nabla f(x_t, \xi_t)\|}\right\}\nabla f(x_t, \xi_t)$ |
| $\theta_t$ | $g_t - \nabla f(x_t)$ |
| $\hat{\theta}_t$ | $g_t - \texttt{clip}(\nabla f(x_t), \lambda/2)$ |
| $\theta_t^u$ | $g_t - \mathbb{E}_{\xi_t}[g_t]$ |
| $\theta_t^b$ | $\mathbb{E}_{\xi_t}[g_t] - \nabla f(x_t)$ |
| $R_t$ | $\|x_t - x^*\|$ |

The next lemma is used to control the bias and variance of the clipped stochastic gradient.

**Lemma 1** (Lemma 5.1 from (Sadiev et al., 2023)). *Let $X$ be a random vector from $\mathbb{R}^d$ and $\widehat{X} = clip(X, \lambda)$. Then, $\left\|\widehat{X} - \mathbb{E}\left[\widehat{X}\right]\right\| \leq 2\lambda$. Moreover, if for some $\sigma \geq 0$ and $\alpha \in (1, 2]$ we have $\mathbb{E}[X] = x \in \mathbb{R}^d$, $\mathbb{E}[\|X - x\|^\alpha] \leq \sigma^\alpha$, and $\|x\| \leq \frac{\lambda}{2}$, then*

$$\left\|\mathbb{E}\left[\widehat{X}\right] - x\right\| \leq \frac{2^\alpha \sigma^\alpha}{\lambda^{\alpha-1}},$$

$$\mathbb{E}\left[\left\|\widehat{X} - x\right\|^2\right] \leq 18\lambda^{2-\alpha}\sigma^\alpha,$$

$$\mathbb{E}\left[\left\|\widehat{X} - \mathbb{E}\left[\widehat{X}\right]\right\|^2\right] \leq 18\lambda^{2-\alpha}\sigma^\alpha.$$

Moreover, our analysis involves sums of martingale-difference sequences, to which Bernstein's inequality can be applied (Bennett, 1962; Dzhaparidze and Van Zanten, 2001; Freedman et al., 1975).

**Lemma 2** (Bernstein's inequality). *Let the sequence of random variables $\{X_i\}_{i\geq 1}$ form a martingale difference sequence, i.e., $\mathbb{E}[X_i \mid X_{i-1}, \ldots, X_1] = 0$ for all $i \geq 1$. Assume that conditional variances $\sigma_i^2 = \mathbb{E}[X_i^2 \mid X_{i-1}, \ldots, X_1]$ exist and are bounded and also assume that there exists deterministic constant $c > 0$ such that $|X_i| \leq c$ almost surely for all $i \geq 1$. Then for all $b > 0$, $G > 0$ and $n \geq 1$*

$$\mathbb{P}\left\{\left|\sum_{i=1}^n X_i\right| > b \text{ and } \sum_{i=1}^n \sigma_i^2 \leq G\right\} \leq 2\exp\left(-\frac{b^2}{2G + \frac{2cb}{3}}\right).$$

Below, we also list some useful properties of Assumption 2.

**Proposition 1** (Gorbunov et al. (2025), Lemma 2.2). *Suppose that Assumption 2 holds. Then,*

$$\nu\|\nabla f(x)\|^2 \leq 2(L_0 + L_1\|\nabla f(x)\|)(f(x) - f^*),$$

*where $\nu$ is the solution of $x\exp(x) = 1$.*

**Proposition 2** (Chen et al. (2023)). *Assumption 2 is equivalent to*

$$\|\nabla f(y) - \nabla f(x)\| \leq (L_0 + L_1\|\nabla f(x)\|)\exp(L_1\|y - x\|)\|y - x\|, \quad \forall x, y \in \mathbb{R}^d.$$

Finally, in the next paragraph, we give an overview of prior convergence results derived for $(L_0, L_1)$-smooth non-convex problems.

**Convergence under $(L_0, L_1)$-smoothness.** Early studies on the convergence of first-order methods under $(L_0, L_1)$-smoothness has primarily focused on the non-convex setting. Zhang et al. (2020b) introduced this smoothness condition and demonstrated that Clip-GD achieves an iteration complexity of $\mathcal{O}\left(\max\left\{L_0\Delta/\varepsilon^2, (1+L_1^2)\Delta/L_0\right\}\right)$ with $\Delta := f(x) - \inf_{x\in\mathbb{R}^d} f(x)$ for finding $\varepsilon$-approximate first-order stationary point. The dominant term in the derived bound is independent

of $L_1$ and can be significantly smaller than the complexity of GD. This foundational work has since been extended to include methods with momentum and clipping (Zhang et al., 2020a), as well as variants like Normalized GD (Zhao et al., 2021; Chen et al., 2023), its momentum-based counterpart (Hübler et al., 2024b), its distributed version with compression (Khirirat et al., 2024), SignGD (Crawshaw et al., 2022), adaptive methods like AdaGrad and Adam (Faw et al., 2023; Wang et al., 2022; 2023; Li et al., 2024), and Armijo-like gradient methods (Bilel, 2024). More recently, Vankov et al. (2025) further improved these results by deriving a tighter complexity bound for Clip-GD of $\mathcal{O}\left(\max\left\{L_0\Delta/\varepsilon^2, L_1\Delta/\varepsilon\right\}\right)$.

## B   MISSING PROOFS

### B.1   LEMMAS

**Lemma 3** (Different cases). *Suppose that Assumptions 1 and 2 hold. Then, the sequence $\{x_k\}_{k=0}^{K}$, generated by Algorithm 1 after $K$ iterations, satisfies following inequalities.*

**Case 1**. *If $\|\nabla f(x_k)\| \leq \frac{L_0}{L_1}$, we have*

$$\gamma(f(x_k) - f^*) \leq \|x_k - x^*\|^2 - \|x_{k+1} - x^*\|^2 - 2\gamma\langle\theta_k, x_k - x^*\rangle + 2\gamma^2\|\theta_k\|^2$$

*with $\theta_k := g_k - \nabla f(x_k)$, $\gamma \leq \frac{1}{16L_0}$ and any $\lambda > 0$.*

**Case 2**. *If $\frac{\lambda}{2} \geq \|\nabla f(x_k)\| \geq \frac{L_0}{L_1}$, then*

$$\gamma(f(x_k) - f^*) \leq \|x_k - x^*\|^2 - \|x_{k+1} - x^*\|^2 - 2\gamma\langle\theta_k, x_k - x^*\rangle + 2\gamma^2\|\theta_k\|^2$$

*with $\theta_k := g_k - \nabla f(x_k)$, $\gamma \leq \frac{1}{8L_1\lambda}$.*

**Case 3**. *If $\|\nabla f(x_k)\| \geq \frac{\lambda}{2} \geq \frac{L_0}{L_1}$, we get*

$$\|x_{k+1} - x^*\|^2 \leq \|x_k - x^*\|^2 - \frac{\gamma\lambda}{16L_1} - 2\gamma\langle\hat{\theta}_k, x_k - x^*\rangle$$

*with $\hat{\theta}_k := g_k - \mathtt{clip}(\nabla f(x_k), \lambda/2)$, $\gamma \leq \frac{1}{16L_1\lambda}$.*

*Proof.* We start our proof using the update rule of Algorithm 1:

$$\|x_{k+1} - x^*\|^2 = \|x_k - x^*\|^2 - 2\gamma\langle g_k, x_k - x^*\rangle + \gamma^2\|g_k\|^2. \tag{7}$$

The rest of the proof depends on the relation between $\lambda$, $\|\nabla f(x_k)\|$, and $\frac{L_0}{L_1}$.

**Case 1:** $\|\nabla f(x_k)\| \leq \frac{L_0}{L_1}$. Using the definition of $\theta_k$ (see Table 2), we can decompose (7) as follows:

$$\|x_{k+1} - x^*\|^2 \leq \|x_k - x^*\|^2 - 2\gamma\langle\nabla f(x_k), x_k - x^*\rangle - 2\gamma\langle\theta_k, x_k - x^*\rangle$$
$$+ 2\gamma^2\|\nabla f(x_k)\|^2 + 2\gamma^2\|\theta_k\|^2. \tag{8}$$

Using Proposition 1 with $\|\nabla f(x_k)\| \leq \frac{L_0}{L_1}$ and $\nu \geq \frac{1}{2}$, we get

$$\|\nabla f(x_k)\|^2 \leq 4(L_0 + L_1\|\nabla f(x_k)\|)(f(x_k) - f^*) \leq 8L_0(f(x_k) - f^*), \tag{9}$$

where we also use $\|\nabla f(x_k)\| \leq \frac{L_0}{L_1}$ in the last step. Applying the convexity of $f$ (Assumption 1) and substituting (9) into (8), one can obtain

$$\|x_{k+1} - x^*\|^2 \leq \|x_k - x^*\|^2 - 2\gamma\langle\theta_k, x_k - x^*\rangle + 2\gamma^2\|\theta_k\|^2 - (2\gamma - 16\gamma^2 L_0)(f(x_k) - f^*).$$

Then, the above inequality combined with the stepsize condition $\gamma \leq \frac{1}{16L_0}$ imply

$$\gamma(f(x_k) - f^*) \leq \|x_k - x^*\|^2 - \|x_{k+1} - x^*\|^2 - 2\gamma\langle\theta_k, x_k - x^*\rangle + 2\gamma^2\|\theta_k\|^2.$$

**Case 2:** $\frac{\lambda}{2} \geq \|\nabla f(x_k)\| > \frac{L_0}{L_1}$. In this case, Proposition 1 gives

$$\|\nabla f(x_k)\|^2 \leq 4(L_0 + L_1\|\nabla f(x_k)\|)(f(x_k) - f^*) \leq 8L_1\|\nabla f(x_k)\|(f(x_k) - f^*). \tag{10}$$

Therefore, using the same decomposition (8) as in **Case 1**, applying (10) and choosing $\gamma \leq \frac{1}{8L_1\lambda}$, we obtain

$$\|x_{k+1} - x^*\|^2 \leq \|x_k - x^*\|^2 - 2\gamma\langle\nabla f(x_k), x_k - x^*\rangle - 2\gamma\langle\theta_k, x_k - x^*\rangle$$
$$+ 2\gamma^2\|\nabla f(x_k)\|^2 + 2\gamma^2\|\theta_k\|^2$$
$$\leq \|x_k - x^*\|^2 - 2\gamma\langle\theta_k, x_k - x^*\rangle + 2\gamma^2\|\theta_k\|^2$$
$$- (2\gamma - 16\gamma^2 L_1\|\nabla f(x_k)\|)(f(x_k) - f^*)$$
$$\leq \|x_k - x^*\|^2 - 2\gamma\langle\theta_k, x_k - x^*\rangle + 2\gamma^2\|\theta_k\|^2$$
$$- (2\gamma - 8\gamma^2 L_1\lambda)(f(x_k) - f^*)$$
$$\leq \|x_k - x^*\|^2 - 2\gamma\langle\theta_k, x_k - x^*\rangle + 2\gamma^2\|\theta_k\|^2 - \gamma(f(x_k) - f^*),$$

where we use $\|\nabla f(x_k)\| \leq \frac{\lambda}{2}$. Rearranging the terms, we conclude this part of the proof.

**Case 3:** $\|\nabla f(x_k)\| > \frac{\lambda}{2} \geq \frac{L_0}{L_1}$. Using this relation and the definition of $\hat{\theta}_k$ (see Table 2), we decompose (7):

$$\|x_{k+1} - x^*\|^2 = \|x_k - x^*\|^2 - 2\gamma\langle g_k, x_k - x^*\rangle + \gamma^2\|g_k\|^2$$

$$\leq \|x_k - x^*\|^2 - \frac{\gamma\lambda}{\|\nabla f(x_k)\|}\langle\nabla f(x_k), x_k - x^*\rangle - 2\gamma\langle\hat{\theta}_k, x_k - x^*\rangle + \gamma^2\lambda^2. \quad (11)$$

Applying the convexity of $f$, and then combining it with (10), one can get

$$\|x_{k+1} - x^*\|^2 \leq \|x_k - x^*\|^2 - \frac{\gamma\lambda}{\|\nabla f(x_k)\|}\langle\nabla f(x_k), x_k - x^*\rangle - 2\gamma\langle\hat{\theta}_k, x_k - x^*\rangle + \gamma^2\lambda^2$$

$$\leq \|x_k - x^*\|^2 - \frac{\gamma\lambda}{\|\nabla f(x_k)\|}(f(x_k) - f^*) - 2\gamma\langle\hat{\theta}_k, x_k - x^*\rangle + \gamma^2\lambda^2$$

$$\overset{(10)}{\leq} \|x_k - x^*\|^2 - \frac{\gamma\lambda}{8L_1} - 2\gamma\langle\hat{\theta}_k, x_k - x^*\rangle + \gamma^2\lambda^2.$$

Using that $\gamma \leq \frac{1}{16L_1\lambda}$, we have

$$\|x_{k+1} - x^*\|^2 \leq \|x_k - x^*\|^2 - \frac{\gamma\lambda}{8L_1} - 2\gamma\langle\hat{\theta}_k, x_k - x^*\rangle + \gamma^2\lambda^2$$

$$\leq \|x_k - x^*\|^2 - \frac{\gamma\lambda}{16L_1} - 2\gamma\langle\hat{\theta}_k, x_k - x^*\rangle.$$

This concludes the proof. $\qquad\square$

**Remark 1.** *We note, the **Case 1** does not use the fact that $\lambda \geq \frac{2L_0}{L_1}$. Moreover, as will be shown later, the proof of the main result has two possible regimes, and for each of them we will apply the corresponding cases from Lemma 3.*

**Lemma 4** (Descent lemma). *Suppose that Assumptions 1 and 2 hold. Then, after $K$ iterations of Algorithm 1, we have two possible options.*

**Option 1.** *If for all $k = 0, \ldots, K - 1$ the inequality $\|\nabla f(x_k)\| \leq \frac{L_0}{L_1}$ holds, $\lambda > 0$, and $\gamma \leq \frac{1}{16L_0}$, then*

$$\sum_{k=0}^{K-1}\gamma(f(x_k) - f^*) \leq \|x_0 - x^*\|^2 - \|x_K - x^*\|^2 - \sum_{k=0}^{K-1}2\gamma\langle\theta_k, x_k - x^*\rangle + \sum_{k=0}^{K-1}\|\theta_k\|^2.$$

**Option 2.** *If $\lambda \geq \frac{2L_0}{L_1}$ and $\gamma \leq \min\left\{\frac{1}{16L_0}, \frac{1}{16L_1\lambda}\right\}$, then*

$$\sum_{k\in T_1\cup T_2}\gamma(f(x_k) - f^*) \leq \|x_0 - x^*\|^2 - \|x_K - x^*\|^2 - \sum_{k\in T_1\cup T_2}2\gamma\langle\theta_k, x_k - x^*\rangle$$

$$+ \sum_{k\in T_1\cup T_2}2\gamma^2\|\theta_k\|^2 - \sum_{k\in T_3}2\gamma\langle\hat{\theta}_k, x_k - x^*\rangle - \frac{\gamma\lambda|T_3|}{16L_1},$$

*where*

$$T_1 := T_1(K) := \left\{k \in 0, \ldots, K - 1 \,\Big|\, \|\nabla f(x_k)\| \leq \frac{L_0}{L_1}\right\},$$

$$T_2 := T_2(K) := \left\{k \in 0, \ldots, K - 1 \,\Big|\, \frac{\lambda}{2} \geq \|\nabla f(x_k)\| > \frac{L_0}{L_1}\right\},$$

$$T_3 := T_3(K) := \left\{k \in 0, \ldots, K - 1 \,\Big|\, \|\nabla f(x_k)\| > \frac{\lambda}{2}\right\}.$$

*Proof.* The final result follows directly from Lemma 3. Specifically, for the first option, we use only *Case 1* from Lemma 3, and for the second one, we apply *Cases 1, 2, 3*, respectively. Hence, aggregating the inequalities established therein yields the desired conclusion and completes the proof. $\quad\square$

**Remark 2.** *It is worth noting that Lemma 4 covers the case of $L_1 = 0$. Indeed, in this case, we have $L_0/L_1 = \infty$, meaning that $\|\nabla f(x_k)\| \leq L_0/L_1$ is always satisfied, i.e., one can consider Option 1 only.*

## B.2 PROOF OF THEOREM 1

**Theorem 2** (Theorem 1). *Let Assumptions 1, 2, and 3 hold. Then, after $K$ iterations of* Clip-SGD *(Algorithm 1) with*

$$\lambda = \max\left\{2L_0 \min\left\{4R_0, \frac{1}{L_1}\right\}, 9^{\frac{1}{\alpha}}\sigma K^{\frac{1}{\alpha}}\left(\ln\left(\frac{4K}{\delta}\right)\right)^{-\frac{1}{\alpha}}\right\}, \tag{12}$$

$$\gamma = \frac{1}{160\lambda \ln\left(\frac{4K}{\delta}\right)} \min\left\{4R_0, \frac{1}{L_1}\right\}, \tag{13}$$

*we have the following result.*

- *If $4R_0 \leq \frac{1}{L_1}$, then*

$$f\left(\frac{1}{K}\sum_{k=0}^{K-1} x_k\right) - f^* = \tilde{\mathcal{O}}\left(\frac{L_0 R_0^2}{K}, \frac{R_0\sigma}{K^{(\alpha-1)/\alpha}}\right)$$

  *with probability at least $1 - \delta$.*

- *If $4R_0 \geq \frac{1}{L_1}$ and $K = \Omega\left(\frac{(L_1 R_0)^{2+\alpha}\ln^2\left(\frac{4K}{\delta}\right)}{\delta}\right)$*

$$\min_{k=0,\ldots,K-1}(f(x_k) - f^*) = \tilde{\mathcal{O}}\left(\max\left\{\frac{L_0 R_0^2}{K}, \frac{L_1 R_0^2\sigma}{K^{(\alpha-1)/\alpha}}\right\}\right)$$

  *with probability at least $1 - 2\delta$.*

*Proof.* The main idea behind the proof lies in the careful analysis of regimes characterized by the relationship between the initial distance to the optimum ($R_0$) and $\frac{1}{L_1}$. To be more precise, we consider two different regimes: $4R_0 \leq 1/L_1$ and $4R_0 \geq 1/L_1$. Using this, we construct the proof as follows.

**Part 1.** First, we decompose Lemma 4 according to the introduced regimes and define "good" probability events $E_k$, implying the desired result.

**Part 2.** Next, we propose unified bounds for the terms from the first part, in both regimes as well.

**Part 3.** The third part is related to the second regime only.

**Part 4.** The fourth part concludes the proof, i.e., we show that $\mathbb{P}\{E_k\}$ is large enough.

**Part 1: Decomposition.**

*Regime 1:* $4R_0 \leq 1/L_1$. The proof for this regime closely follows the proof of Theorem E.6 from Sadiev et al. (2023). Let us denote the probabilistic event $E_k$: the inequalities

$$-\sum_{l=0}^{t-1} 2\gamma\langle\theta_l, x_l - x^*\rangle + \sum_{l=0}^{t-1} 2\gamma^2\|\theta_l\|^2 \leq R_0^2,$$

$$R_k \leq \sqrt{2}R_0$$

hold simultaneously for $t = 0, \ldots, k$. We want to show via induction that $\mathbb{P}\{E_k\} \geq 1 - \frac{k\delta}{K}$. The case of $k = 0$ is obvious. Then, let us assume that the event $E_{T-1}$ with $T \leq K$ holds with the probability $\mathbb{P}\{E_{T-1}\} \geq 1 - \frac{(T-1)\delta}{K}$. Also this event implies that $x_t \in B_{\sqrt{2}R_0}(x^*)$ for all $t = 0, \ldots, T-1$. Therefore, we have that $E_{T-1}$ implies

$$\|x_T - x^*\| = \|x_{T-1} - \gamma g_T - x^*\| \leq \|x_{T-1} - x^*\| + \gamma\lambda \overset{(13)}{\leq} 2R_0.$$

Consequently, $\{x_t\}_{t=0}^T \subseteq B_{2R_0}(x^*)$ follows from $E_{T-1}$. Therefore, the event $E_{T-1}$ implies

$$\|\nabla f(x_t)\| \overset{\text{Prop. }2}{\leq} L_0 R_t \exp(L_1 R_t) \leq \sqrt{2} L_0 R_0 \exp\left(\sqrt{2} L_1 R_0\right) \leq 4 L_0 R_0 \leq \frac{L_0}{L_1} \quad (14)$$

for all $t = 0, \dots, T-1$. Thus, we can apply Lemma 4 (*Option 1*): $E_{T-1}$ implies that

$$\sum_{l=0}^{t-1} \gamma(f(x_l) - f^*) \leq \|x_0 - x^*\|^2 - \|x_t - x^*\|^2 - \sum_{l=0}^{t-1} 2\gamma\langle\theta_l, x_l - x^*\rangle + \sum_{l=0}^{t-1} 2\gamma^2\|\theta_l\|^2 \quad (15)$$

holds for $t = 1, \dots, T$. It is worth mentioning that (12) and (13) give $\gamma \leq \frac{1}{16 L_0}$. What is more, event $E_{T-1}$ implies that

$$\sum_{l=0}^{t-1} \gamma(f(x_l) - f^*) \leq \|x_0 - x^*\|^2 - \|x_t - x^*\|^2 - \sum_{l=0}^{t-1} 2\gamma\langle\theta_l, x_l - x^*\rangle + \sum_{l=0}^{t-1} 2\gamma^2\|\theta_l\|^2 \leq 2R_0^2$$

for $t = 1, \dots, T-1$. Moreover, the bound $f(x_l) - f^* \geq 0$ with (15) leads to

$$R_T^2 \leq R_0^2 - \sum_{t=0}^{T-1} 2\gamma\langle\theta_t, x_t - x^*\rangle + \sum_{t=0}^{T-1} 2\gamma^2\|\theta_t\|^2. \quad (16)$$

Next, we define random vectors

$$\eta_t = \begin{cases} x_t - x^*, & \|x_t - x^*\| \leq \sqrt{2}R_0, \\ 0, & \text{otherwise,} \end{cases}$$

for all $t = 0, \dots, T-1$. According to the definition, $\eta_t$ is bounded with probability 1:

$$\|\eta_t\| \leq \sqrt{2}R_0.$$

Moreover, the event $E_{T-1}$ implies $\|x_t - x^*\| \leq \sqrt{2}R_0$ for all $t = 0, \dots, T-1$. As a result, we get $\eta_t = x_t - x^*$ within this event. Now let us decompose (16) using the notation of $\theta_t^u$, $\theta_t^b$ and $\eta_t$:

$$\|x_T - x^*\|^2 \leq R_0^2 - \underbrace{\sum_{t \in T_1(T) \cup T_2(T)} 2\gamma\langle\theta_t^u, \eta_t\rangle}_{①} - \underbrace{\sum_{t \in T_1(T) \cup T_2(T)} 2\gamma\langle\theta_t^b, \eta_t\rangle}_{②}$$

$$+ \underbrace{\sum_{t \in T_1(T) \cup T_2(T)} 4\gamma^2\left[\|\theta_t^u\|^2 - \mathbb{E}_{\xi_t}\left[\|\theta_t^u\|^2\right]\right]}_{③} + \underbrace{\sum_{t \in T_1(T) \cup T_2(T)} 4\gamma^2\mathbb{E}_{\xi_t}\left[\|\theta_t^u\|^2\right]}_{④}$$

$$+ \underbrace{\sum_{t \in T_1(T) \cup T_2(T)} 4\gamma^2\|\theta_t^b\|^2}_{⑤}, \quad (17)$$

where we also use the definitions of $T_1(T), T_2(T)$, and $T_3(T)$, and the fact that in this regime $T_3(T) \equiv 0$ for any $T \geq 0$.

**Regime 2:** $4R_0 \geq 1/L_1$ Similarly to the first regime, let us denote the probabilistic event $E_k$: the inequalities

$$-\sum_{l \in T_1(t) \cup T_2(t)} 2\gamma\langle\theta_l, x_l - x^*\rangle + \sum_{l \in T_1(t) \cup T_2(t)} 2\gamma^2\|\theta_l\|^2 \leq R_0^2,$$

$$-\sum_{l \in T_3(t)} 2\gamma\langle\hat{\theta}_l, x_l - x^*\rangle \leq \frac{\gamma\lambda|T_3(t)|}{32 L_1},$$

$$R_k \leq \sqrt{2}R_0$$

hold simultaneously for $t = 0, \ldots, k$. If the first and third inequalities coincide with the previous case, the second inequality is also necessary for our analysis. We want to show via induction that $\mathbb{P}\{E_k\} \geq 1 - \frac{k\delta}{K} - \sum_{r=0}^{k} \min\{r, C_1\}\delta_0 \mathbb{P}\{|T_3(k)| = r\}$, where $C_1$ and $\delta_0$ will be defined later. The case of $k = 0$ is obvious. Then, let us assume that the event $E_{T-1}$ with $T \leq K$ holds with probability $\mathbb{P}\{E_{T-1}\} \geq 1 - \frac{(T-1)\delta}{K} - \sum_{r=0}^{T-1} \min\{r, C_1\}\delta_0 \mathbb{P}\{|T_3(T-1)| = r\}$. Also this event implies that $x_t \in B_{\sqrt{2}R_0}(x^*)$ for all $t = 0, \ldots, T - 1$. Therefore, we have

$$\|x_T - x^*\| = \|x_{T-1} - \gamma g_T - x^*\| \leq \|x_{T-1} - x^*\| + \gamma\lambda \overset{(13)}{\leq} 2R_0$$

within event $E_{T-1}$. Consequently, $\{x_t\}_{t=0}^{T} \subseteq B_{2R_0}(x^*)$ follows from $E_{T-1}$, and we can apply Lemma 4 (*Option* 2): event $E_{T-1}$ implies that inequality

$$\sum_{l \in T_1(t) \cup T_2(t)} \gamma(f(x_l) - f^*) \leq \|x_0 - x^*\|^2 - \|x_t - x^*\|^2 - \sum_{l \in T_1(t) \cup T_2(t)} 2\gamma\langle\theta_l, x_l - x^*\rangle$$

$$+ \sum_{l \in T_1(t) \cup T_2(t)} 2\gamma^2\|\theta_l\|^2 - \sum_{l \in T_3(t)} 2\gamma\langle\hat{\theta}_l, x_l - x^*\rangle - \frac{\gamma\lambda|T_3(t)|}{16L_1} \quad (18)$$

holds for $t = 1, \ldots, T$. Also let us clarify that $\gamma \leq \min\left\{\frac{1}{16L_0}, \frac{1}{16L_1\lambda}\right\}$ due to (13). What is more, the event $E_{T-1}$ implies that

$$\sum_{l \in T_1(t) \cup T_2(t)} \gamma(f(x_l) - f^*) \leq \|x_0 - x^*\|^2 - \|x_t - x^*\|^2 - \sum_{l \in T_1(t) \cup T_2(t)} 2\gamma\langle\theta_l, x_l - x^*\rangle$$

$$+ \sum_{l \in T_1(t) \cup T_2(t)} 2\gamma^2\|\theta_l\|^2 - \sum_{l \in T_3(t)} 2\gamma\langle\hat{\theta}_l, x_l - x^*\rangle - \frac{\gamma\lambda|T_3(t)|}{16L_1}$$

$$\leq 2R_0^2 - \frac{\gamma\lambda|T_3(t)|}{32L_1}$$

for $t = 1, \ldots, T - 1$. Moreover, $E_{T-1}$ with $f(x_t) - f^* \geq 0$ implies

$$0 \leq 2R_0^2 - \frac{\gamma\lambda|T_3(T-1)|}{32L_1} \Rightarrow |T_3(T-1)| \leq 64 \cdot 160(L_1R_0)^2 \ln\left(\frac{4K}{\delta}\right)$$

due to (13). Therefore, the events $E_k$ and $E_k \cap \{|T_3(k)| \leq C_1 := 64 \cdot 160(L_1R_0)^2 \ln\left(\frac{4K}{\delta}\right)\}$ are *equal*. What is more, from (18) we have

$$\|x_T - x^*\|^2 \leq R^2 - \sum_{t \in T_1(T) \cup T_2(T)} 2\gamma\langle\theta_t, x_t - x^*\rangle + \sum_{t \in T_1(T) \cup T_2(T)} 2\gamma^2\|\theta_t\|^2$$

$$- \sum_{t \in T_3(T)} 2\gamma\langle\hat{\theta}_t, x_t - x^*\rangle - \frac{\gamma\lambda|T_3(T)|}{16L_1} \quad (19)$$

within event $E_{T-1}$. Next, we define random vectors

$$\eta_t = \begin{cases} x_t - x^*, & \|x_t - x^*\| \leq \sqrt{2}R_0, \\ 0, & \text{otherwise}, \end{cases}$$

for all $t = 0, \ldots, T - 1$. According to the definition, $\eta_t$ is bounded with probability 1:

$$\|\eta_t\| \leq \sqrt{2}R_0.$$

Moreover, the event $E_{T-1}$ implies $\|x_t - x^*\| \leq \sqrt{2}R_0$ for all $t = 0, \ldots, T-1$. As a result, we get $\eta_t = x_t - x^*$ within this event. Now let us decompose (19) using the notation of $\theta_t^u, \theta_t^b$ and $\eta_t$:

$$
R_T^2 \leq R_0^2 - \underbrace{\sum_{t \in T_1(T) \cup T_2(T)} 2\gamma\langle\theta_t^u, \eta_t\rangle}_{\text{①}} - \underbrace{\sum_{t \in T_1(T) \cup T_2(T)} 2\gamma\langle\theta_t^b, \eta_t\rangle}_{\text{②}}
$$

$$
+ \underbrace{\sum_{t \in T_1(T) \cup T_2(T)} 4\gamma^2\left[\|\theta_t^u\|^2 - \mathbb{E}_{\xi_t}\left[\|\theta_t^u\|^2\right]\right]}_{\text{③}} + \underbrace{\sum_{t \in T_1(T) \cup T_2(T)} 4\gamma^2\mathbb{E}_{\xi_t}\left[\|\theta_t^u\|^2\right]}_{\text{④}}
$$

$$
+ \underbrace{\sum_{t \in T_1(T) \cup T_2(T)} 4\gamma^2\|\theta_t^b\|^2}_{\text{⑤}} - \underbrace{\sum_{t \in T_3(T)} 2\gamma\langle\hat{\theta}_t, \eta_t\rangle}_{\text{⑥}} - \frac{\gamma\lambda|T_3(T)|}{16L_1}. \tag{20}
$$

**Part 2: Bounds for ① − ⑤.**

In this part of the proof, we can bound terms ① − ⑤ from (17) and (20). For *Regime 1*, it is worth mentioning that event $E_{T-1}$ implies

$$
T_1(T) = \{0, \ldots, T-1\} \tag{21}
$$

due to (14). What is more, according to (12), we have

$$
\|\nabla f(x_t)\| \leq 4L_0R_0 \leq \frac{\lambda}{2}
$$

for all $t = 0, \ldots, T-1$ within the event $E_{T-1}$. Considering the second regime ($4R_0 \geq 1/L_1$), by definition of $T_i(T)$ from Lemma 4, we have that for all $t \in T_1(T) \cup T_2(T)$

$$
\|\nabla f(x_t)\| \leq \frac{\lambda}{2}.
$$

Consequently, using (21) for the case $4R_0 \leq 1/L_1$, we will bound terms ① − ⑤ in the unified form. To continue, we can apply Lemma 1 to obtain that

$$
\|\theta_t^u\| \leq 2\lambda, \tag{22}
$$

$$
\|\theta_t^b\| \leq \frac{2^\alpha\sigma^\alpha}{\lambda^{\alpha-1}}, \tag{23}
$$

$$
\mathbb{E}_{\xi_t}\left[\|\theta_t^u\|^2\right] \leq 18\lambda^{2-\alpha}\sigma^\alpha \tag{24}
$$

for all $t \in T_1(T) \cup T_2(T)$. Hence, we can apply (22), (23) and (24) to construct bounds for ① − ⑤.

*Upper bound for ①.* First of all, we have

$$
\mathbb{E}_{\xi_t}\left[-2\gamma\langle\theta_t^u, \eta_t\rangle\right] = 0,
$$

since $\mathbb{E}_{\xi_t}[\,\cdot\,] = \mathbb{E}_{\xi_t}[\,\cdot\,|\xi_{t-1}, \xi_{t-2}, \ldots]$, $\mathbb{E}_{\xi_t}[\eta_t] = \eta_t$, and $\mathbb{E}_{\xi_t}[\theta_t^u] = 0$. Moreover,

$$
|-2\gamma\langle\theta_t^u, \eta_t\rangle| \leq 2\gamma\|\theta_t^u\|\|\eta_t\| \overset{(22)}{\leq} 6\gamma\lambda R_0 \overset{(13)}{\leq} \frac{3R_0^2}{20\ln\left(\frac{4K}{\delta}\right)} = c.
$$

What is more, let us define $\sigma_t^2 = \mathbb{E}\left[4\gamma^2\langle\theta_t^u, \eta_t\rangle^2\right]$. Then, we get

$$
\sigma_t^2 \leq \mathbb{E}_{\xi_t}\left[4\gamma^2\|\theta_t^u\|^2\|\eta_t\|^2\right] \leq 8\gamma^2 R_0^2\mathbb{E}_{\xi_t}\left[\|\theta_t^u\|^2\right].
$$

As a consequence, we can apply Bernstein's inequality with $b = \frac{R_0^2}{5}$ and $G = \frac{R_0^4}{100\ln\left(\frac{4K}{\delta}\right)}$:

$$
\mathbb{P}\left\{|\text{①}| > b \text{ and } \sum_{t \in T_1(T) \cup T_2(T)} \sigma_t^2 \leq G\right\} \leq 2\exp\left(-\frac{b^2}{2G + 2cb/3}\right) = \frac{\delta}{2K}.
$$

Thus, we get

$$\mathbb{P}\left\{|\text{①}| \leq b \text{ or } \sum_{t \in T_1(T) \cup T_2(T)} \sigma_t^2 > G\right\} \geq 1 - \frac{\delta}{2K}.$$

Moreover, the event $E_{T-1}$ implies

$$\sum_{t \in T_1(T) \cup T_2(T)} \sigma_t^2 \leq \sum_{t \in T_1(T) \cup T_2(T)} 8\gamma^2 R_0^2 \mathbb{E}_{\xi_t}\left[\|\theta_t^u\|^2\right] \stackrel{(24)}{\leq} 144\gamma^2\lambda^{2-\alpha}\sigma^\alpha R_0^2(|T_1(T)| + |T_2(T)|)$$

$$\leq 144\gamma^2\lambda^{2-\alpha}\sigma^\alpha R_0^2 K = \frac{144\gamma^2\lambda^2\sigma^\alpha R_0^2 K}{\lambda^\alpha} \stackrel{(12)}{\leq} 16\gamma^2\lambda^2 R_0^2 \ln\left(\frac{4K}{\delta}\right)$$

$$\stackrel{(13)}{\leq} \frac{R_0^4}{100\ln\left(\frac{4K}{\delta}\right)} = G.$$

*Upper bound for ②.* From the event $E_{T-1}$ it follows that

$$-\sum_{t \in T_1(T) \cup T_2(T)} 2\gamma\langle\theta_t^b, \eta_t\rangle \leq \sum_{t \in T_1(T) \cup T_2(T)} 2\gamma\|\theta_t^b\|\|\eta_t\| \stackrel{(23)}{\leq} \frac{4 \cdot 2^\alpha\gamma\sigma^\alpha R_0 K}{\lambda^{\alpha-1}}$$

$$= \frac{4 \cdot 2^\alpha\gamma\lambda\sigma^\alpha R_0 K}{\lambda^\alpha} \stackrel{(12),(13)}{\leq} \frac{16R_0^2}{360} \leq \frac{R_0^2}{5}.$$

*Upper bound for ③.* We bound ③ in the same way as ①. First, we get

$$\mathbb{E}_{\xi_t}\left[4\gamma^2\left[\|\theta_t^u\|^2 - \mathbb{E}_{\xi_t}\left[\|\theta_t^u\|^2\right]\right]\right] = 0.$$

In addition, we have

$$\left|4\gamma^2\left[\|\theta_t^u\|^2 - \mathbb{E}_{\xi_t}\left[\|\theta_t^u\|^2\right]\right]\right| \stackrel{(22)}{\leq} 32\gamma^2\lambda^2 \stackrel{(12),(13)}{\leq} \frac{R_0^2}{50\ln\left(\frac{4K}{\delta}\right)} = c.$$

Also let us define $\hat{\sigma}_t^2 = \mathbb{E}_{\xi_t}\left[16\gamma^4\left(\|\theta_t^u\|^2 - \mathbb{E}_{\xi_t}\left[\|\theta_t^u\|^2\right]\right)^2\right]$. Thus, one can obtain

$$\hat{\sigma}_t^2 \leq c\mathbb{E}_{\xi_t}\left|4\gamma^2\left[\|\theta_t^u\|^2 - \mathbb{E}_{\xi_t}\left[\|\theta_t^u\|^2\right]\right]\right| \leq 8c\gamma^2\mathbb{E}_{\xi_t}\left[\|\theta_t^u\|^2\right].$$

Consequently, we can apply Bernstein's inequality with $b = \frac{R_0^2}{5}$ and $G = \frac{cR_0^2}{100}$:

$$\mathbb{P}\left\{|\text{③}| > b \quad \text{and} \quad \sum_{t \in T_1(T) \cup T_2(T)} \hat{\sigma}_t^2 \leq G\right\} \leq 2\exp\left(-\frac{b^2}{2G + \frac{2cb}{3}}\right) \leq \frac{\delta}{2K}.$$

At the same time,

$$\mathbb{P}\left\{|\text{③}| \leq b \quad \text{or} \quad \sum_{t \in T_1(T) \cup T_2(T)} \hat{\sigma}_t^2 > G\right\} \geq 1 - \frac{\delta}{2K}.$$

Moreover, the event $E_{T-1}$ implies

$$\sum_{t \in T_1(T) \cup T_2(T)} \hat{\sigma}_t^2 \leq \sum_{t \in T_1(T) \cup T_2(T)} 8c\gamma^2\mathbb{E}_{\xi_t}\left[\|\theta_t^u\|^2\right] \stackrel{(24)}{\leq} \frac{144c\gamma^2\lambda^2\sigma^\alpha K}{\lambda^\alpha} \stackrel{(12)}{\leq} 16c\gamma^2\lambda^2 \ln\left(\frac{4K}{\delta}\right)$$

$$\stackrel{(13)}{\leq} \frac{cR_0^2}{100} = G.$$

*Upper bound for ④.* From $E_{T-1}$ it follows that

$$\sum_{t \in T_1(T) \cup T_2(T)} 4\gamma^2\mathbb{E}_{\xi_t}\left[\|\theta_t^u\|^2\right] \stackrel{(24)}{\leq} 72\gamma^2\lambda^{2-\alpha}\sigma^\alpha K = \frac{72\gamma^2\lambda^2\sigma^\alpha K}{\lambda^\alpha} \stackrel{(12)}{\leq} 8\gamma^2\lambda^2 \ln\left(\frac{4K}{\delta}\right)$$

$$\stackrel{(13)}{\leq} \frac{R_0^2}{200} \leq \frac{R_0^2}{5}.$$

*Upper bound for* ⑤. The event $E_{T-1}$ implies

$$\sum_{t \in T_1(T) \cup T_2(T)} 4\gamma^2 \|\theta_t^b\|^2 \overset{(23)}{\leq} \frac{4 \cdot 2^{2\alpha}\gamma^2\sigma^{2\alpha}K}{\lambda^{2\alpha-2}} \leq \frac{64\gamma^2\lambda^2\sigma^{2\alpha}K}{\lambda^{2\alpha}} \overset{(12)}{\leq} \gamma^2\lambda^2\ln^2\left(\frac{4K}{\delta}\right) \overset{(13)}{\leq} \frac{R_0^2}{5}.$$

### Part 3: Bound of ⑥.

This part is needed only for the second regime since for the first regime $|T_3(T)| = 0$. The main idea lies in the careful decomposition of each term from ⑥. In the deterministic case, it was shown that $|T_3|$ is bounded by a constant (Gorbunov et al., 2025). We aim to achieve a similar effect using just Markov's inequality. We start with the reformulation of each term from ⑥ with the notation of $\hat{\theta}_t$ (see Table 2):

$$\langle \hat{\theta}_t, \eta_t \rangle = \min\left\{1, \frac{\lambda}{\|\nabla f(x_t) + \xi_t\|}\right\} \langle \nabla f(x_t) + \xi_t, \eta_t \rangle$$

$$- \min\left\{1, \frac{\lambda}{2\|\nabla f(x_t)\|}\right\} \langle \nabla f(x_t), \eta_t \rangle$$

$$= \left[\min\left\{1, \frac{\lambda}{\|\nabla f(x_t) + \xi_t\|}\right\} - \min\left\{1, \frac{\lambda}{2\|\nabla f(x_t)\|}\right\}\right] \langle \nabla f(x_t), \eta_t \rangle$$

$$- \min\left\{1, \frac{\lambda}{\|\nabla f(x_t) + \xi_t\|}\right\} \langle \xi_t, \eta_t \rangle$$

$$= \left[\min\left\{1, \frac{\lambda}{\|\nabla f(x_t) + \xi_t\|}\right\} - \frac{\lambda}{2\|\nabla f(x_t)\|}\right] \langle \nabla f(x_t), \eta_t \rangle$$

$$- \min\left\{1, \frac{\lambda}{\|\nabla f(x_t) + \xi_t\|}\right\} \langle \xi_t, \eta_t \rangle, \tag{25}$$

where in the last equation we use $t \in T_3(T)$. Next, let us consider some $B$ such that $0 < B \leq \frac{\lambda}{2}$. We have

$$\mathbb{P}\left\{\min\left\{1, \frac{\lambda}{\|\nabla f(x_t) + \xi_t\|}\right\} - \frac{\lambda}{2\|\nabla f(x_t)\|} \geq 0\right\} = \mathbb{P}\left\{\frac{\lambda\|\nabla f(x_k)\|}{\|\nabla f(x_t) + \xi_t\|} \geq \frac{\lambda}{2}\right\}$$

$$= \mathbb{P}\left\{2\|\nabla f(x_k)\| \geq \|\nabla f(x_t) + \xi_t\|\right\}$$

$$\geq \mathbb{P}\left\{2\|\nabla f(x_k)\| \geq \|\nabla f(x_t)\| + \|\xi_t\|\right\}$$

$$\geq \mathbb{P}\left\{\|\xi_t\| \leq \frac{\lambda}{2}\right\}$$

$$\geq \mathbb{P}\left\{\|\xi_t\| \leq B\right\},$$

where we also use that $\|\nabla f(x_k)\| \geq \frac{\lambda}{2}$. Moreover,

$$\mathbb{P}\left\{\|\xi_t\| \leq B\right\} = \mathbb{P}\left\{\|\xi_t\|^\alpha \leq B^\alpha\right\} \geq 1 - \frac{\sigma^\alpha}{B^\alpha}$$

due to the Markov's inequality. Therefore, using (25), we obtain that

$$-2\gamma\langle \hat{\theta}_t, \eta_t \rangle = -\left[\min\left\{1, \frac{\lambda}{\|\nabla f(x_t) + \xi_t\|}\right\} - \frac{\lambda}{2\|\nabla f(x_t)\|}\right] \langle \nabla f(x_t), x_t - x^* \rangle$$

$$+ 2\gamma\min\left\{1, \frac{\lambda}{\|\nabla f(x_t) + \xi_t\|}\right\} \langle \xi_t, x_t - x^* \rangle$$

$$\leq 2\gamma\|\xi_t\|\|\eta_t\|,$$

where we use the convexity of $f$, identity $\eta_t = x_t - x^*$ within the event $E_{T-1}$, and inequality $\|\xi_t\| \leq B$. What is more, the event $E_{T-1}$ with $\|\xi_t\| \leq B$ implies

$$2\gamma\|\xi_t\|\|\eta_t\| \leq 4\gamma B R_0.$$

Choosing $B = \frac{\lambda}{128L_1R_0}$, we finally have in this case

$$-2\gamma\langle \hat{\theta}_t, x_t - x^* \rangle \leq \frac{\gamma\lambda}{32L_1}.$$

**Part 4: Final bound.**

In this part, we combine the derived bounds and estimate the probability of $E_T$. First, let us denote

$$E_{①} = \left\{ |①| \leq \frac{R_0^2}{5} \text{ or } \sum_{t \in T_1(T) \cup T_2(T)} \sigma_t^2 > \frac{R_0^4}{100 \ln\left(\frac{4K}{\delta}\right)} \right\},$$

$$E_{③} = \left\{ |③| \leq \frac{R_0^2}{5} \text{ or } \sum_{t \in T_1(T) \cup T_2(T)} \sigma_t^2 > \frac{R_0^4}{5000 \ln\left(\frac{4K}{\delta}\right)} \right\},$$

$$E_{\text{Markov}} = \left\{ \|\xi_{T-1}\| \leq B \text{ or } (T-1) \notin T_3(T) \text{ or } |T_3(T-1)| > C_1 - 1 \right\}.$$

According to the parts $1, 2$ and $3$, we obtain

$$\mathbb{P}\left\{ E_{T-1} \right\} \geq 1 - \frac{(T-1)\delta}{K} - \sum_{r=0}^{T-1} \min\{r, C_1\} \mathbb{P}\{|T_3(T-1)| = r\},$$

$$\mathbb{P}\left\{ E_{①} \right\} \geq 1 - \frac{\delta}{2K},$$

$$\mathbb{P}\left\{ E_{③} \right\} \geq 1 - \frac{\delta}{2K},$$

$$\mathbb{P}\left\{ \overline{E}_{\text{Markov}} \right\} \leq \frac{128^\alpha (L_1 R_0)^\alpha \sigma^\alpha}{\lambda^\alpha} \cdot \mathbb{P}\left\{ (T-1) \in T_3(T) \text{ and } |T_3(T-1)| \leq C_1 - 1 \right\}$$

with $C_1 = 64 \cdot 160 (L_1 R_0)^2 \ln\left(\frac{4K}{\delta}\right)$. Moreover, we have

$$1 - \frac{128^\alpha (L_1 R_0)^\alpha \sigma^\alpha}{\lambda^\alpha} \geq 1 - \frac{128^\alpha (L_1 R_0)^\alpha \sigma^\alpha \ln\left(\frac{4K}{\delta}\right)}{9\sigma^\alpha K} = 1 - \frac{128^\alpha (L_1 R_0)^\alpha \ln\left(\frac{4K}{\delta}\right)}{9K} \geq 1 - \delta_0,$$

with $K \geq \frac{128^\alpha \ln\left(\frac{4K}{\delta}\right)(L_1 R_0)^\alpha}{9\delta_0}$. Next, we consider again two possible regimes.

*Regime 1:* $4R_0 \leq {}^1/_{L_1}$. **Part 3** is not needed in this regime. Thus, we get that $E_{T-1} \cap E_{①} \cap E_{③}$ implies

$$R_T^2 \leq R_0^2 + \frac{R_0^2}{5} + \frac{R_0^2}{5} + \frac{R_0^2}{5} + \frac{R_0^2}{5} + \frac{R_0^2}{5} = 2R_0^2,$$

which also guarantees that the event $E_T$ holds. Thus, we get

$$\mathbb{P}\{E_T\} \geq \mathbb{P}\{E_{T-1} \cap E_{①} \cap E_{③}\} \geq 1 - \mathbb{P}\{\overline{E}_{T-1}\} - \mathbb{P}\{\overline{E}_{①}\} - \mathbb{P}\{\overline{E}_{③}\} \geq 1 - \frac{T\delta}{K}.$$

This finishes the inductive proof for $4R_0 \leq {}^1/_{L_1}$. In particular, if $T = K$, $E_K$ implies

$$\frac{\sum_{k=0}^{K-1} (f(x_k) - f^*)}{K} \leq \frac{2R_0^2}{\gamma K}.$$

Substituting (13) in the inequality above, noting that we are in the first regime ($4R_0 \leq \frac{1}{L_1}$), and applying Jensen's inequality to the LHS, we conclude that

$$f\left( \frac{1}{K} \sum_{k=0}^{K-1} x_k \right) - f^* = \tilde{\mathcal{O}}\left( \max\left\{ \frac{L_0 R_0^2}{K}, \frac{R_0 \sigma}{K^{1 - \frac{1}{\alpha}}} \right\} \right) \tag{26}$$

with probability at least $1 - \delta$.

*Regime 2:* $4R_0 \geq \frac{1}{L_1}$. First, $E_{T-1}$ with $E_{\text{Markov}}$ implies

$$- \sum_{t \in T_3(T)} 2\gamma \langle \hat{\theta}_t, \eta_t \rangle = - \sum_{t \in T_3(T-1)} 2\gamma \langle \hat{\theta}_t, \eta_t \rangle - 2\gamma \langle \hat{\theta}_{T-1}, \eta_{T-1} \rangle \mathbb{I}\{T - 1 \in T_3(T)\} \leq \frac{\gamma \lambda |T_3(T)|}{32 L_1},$$

where we also use that $T_3(T-1) \subseteq T_3(T)$. Hence, the probability event $E_{T-1} \cap E_① \cap E_③ \cap E_{\text{Markov}}$ implies

$$R_T^2 \leq R_0^2 + R_0^2 - \frac{\gamma\lambda|T_3(T)|}{32L_1},$$

which also guarantees that the event $E_T$ holds. Thus, we get

$$\mathbb{P}\{E_T\} \geq \mathbb{P}\{E_{T-1} \cap E_① \cap E_③ \cap E_{\text{Markov}}\} \geq 1 - \mathbb{P}\{\overline{E}_{T-1}\} - \mathbb{P}\{\overline{E}_①\} - \mathbb{P}\{\overline{E}_③\} - \mathbb{P}\{\overline{E}_{\text{Markov}}\}$$

$$\geq 1 - \frac{T\delta}{K} - \sum_{r=0}^{T-1} \min\{r, C_1\}\delta_0\mathbb{P}\{|T_3(T-1)| = r\}$$

$$- \delta_0\mathbb{P}\{T-1 \in T_3(T) \text{ and } |T_3(T-1)| \leq C_1 - 1\},$$

where the last term comes from the event $\overline{E}_{\text{Markov}}$. Next, let us consider the last two terms in the RHS of the inequality above. First, we introduce events

$$X := \{T-1 \in T_3(T) \text{ and } |T_3(T-1)| \leq C_1 - 1\},$$
$$Y := \{T-1 \in T_3(T) \text{ and } |T_3(T-1)| \geq C_1\},$$
$$Z := \{T-1 \notin T_3(T)\}.$$

Therefore, we get

$$\mathbb{P}\{|T_3(T-1)| = r\} = \mathbb{P}\{|T_3(T-1)| = r|X\}\mathbb{P}\{X\}$$
$$+ \mathbb{P}\{|T_3(T-1)| = r|Y\}\mathbb{P}\{Y\}$$
$$+ \mathbb{P}\{|T_3(T-1)| = r|Z\}\mathbb{P}\{Z\}$$

and

$$\mathbb{P}\{T-1 \in T_3(T) \text{ and } |T_3(T-1)| \leq C_1 - 1\} = \mathbb{P}\{T-1 \in T_3(T) \text{ and } |T_3(T-1)| \leq C_1 - 1|X\}\mathbb{P}\{X\}$$
$$+ \mathbb{P}\{T-1 \in T_3(T) \text{ and } |T_3(T-1)| \leq C_1 - 1|Y\}\mathbb{P}\{Y\}$$
$$+ \mathbb{P}\{T-1 \in T_3(T) \text{ and } |T_3(T-1)| \leq C_1 - 1|Z\}\mathbb{P}\{Z\}.$$

Next, we consider conditional probabilities with respect to $X, Y, Z$. According to the definition of $X$, we have

$$\sum_{r=0}^{T-1} \min\{r, C_1\}\delta_0\mathbb{P}\{|T_3(T-1)| = r|X\} + \delta_0\mathbb{P}\{T-1 \in T_3(T) \text{ and } |T_3(T-1)| \leq C_1 - 1|X\}$$

$$= \sum_{r=0}^{C_1-1} r\delta_0\mathbb{P}\{|T_3(T-1)| = r|X\} + \delta_0$$

$$= \sum_{r=0}^{C_1-1} r\delta_0\mathbb{P}\{|T_3(T)| = r+1|X\} + \delta_0,$$

where the first equation comes from $\mathbb{P}\{|T_3(T-1)| = r|X\} = 0$ for all $r = C_1, \ldots T-1$, and the second equation holds due to $\mathbb{P}\{A|B\} = \mathbb{P}\{A \cap B|B\}$. What is more,

$$\sum_{r=0}^{C_1-1} \mathbb{P}\{|T_3(T)| = r+1|X\} = 1$$

since $|T_3(T)|$ with respect to $X$ can be equal only to $1, \dots, C_1$. Thus, we get

$$\sum_{r=0}^{T-1} \min\{r, C_1\} \delta_0 \mathbb{P}\{|T_3(T-1)| = r | X\} + \delta_0 \mathbb{P}\{T-1 \in T_3(T) \text{ and } |T_3(T-1)| \leq C_1 - 1 | X\}$$

$$= \sum_{r=0}^{C_1-1} r \delta_0 \mathbb{P}\{|T_3(T)| = r + 1 | X\} + \delta_0$$

$$= \sum_{r=0}^{C_1-1} r \delta_0 \mathbb{P}\{|T_3(T)| = r + 1 | X\} + \sum_{r=0}^{C_1-1} \delta_0 \mathbb{P}\{|T_3(T)| = r + 1 | X\}$$

$$= \sum_{r=0}^{C_1-1} (r+1) \delta_0 \mathbb{P}\{|T_3(T)| = r + 1 | X\}$$

$$= \sum_{r=1}^{C_1} r \delta_0 \mathbb{P}\{|T_3(T)| = r | X\}$$

$$= \sum_{r=0}^{T} \min\{r, C_1\} \delta_0 \mathbb{P}\{|T_3(T)| = r | X\}, \tag{27}$$

where in the last equation we add extra zeros for $r = 0$ and $r \geq C_1 + 1$. For event $Y$, we obtain

$$\sum_{r=0}^{T-1} \min\{r, C_1\} \delta_0 \mathbb{P}\{|T_3(T-1)| = r | Y\} + \delta_0 \mathbb{P}\{T-1 \in T_3(T) \text{ and } |T_3(T-1)| \leq C_1 - 1 | Y\}$$

$$= \sum_{r=C_1}^{T-1} C_1 \delta_0 \mathbb{P}\{|T_3(T-1)| = r | Y\}$$

$$= \sum_{r=C_1}^{T-1} C_1 \delta_0 \mathbb{P}\{|T_3(T)| = r + 1 | Y\},$$

where in the first equation we apply the notation of $Y$, and in the second inequality we use that $\mathbb{P}\{A|B\} = \mathbb{P}\{A \cap B|B\}$. Consequently, we get

$$\sum_{r=0}^{T-1} \min\{r, C_1\} \delta_0 \mathbb{P}\{|T_3(T-1)| = r | Y\} + \delta_0 \mathbb{P}\{T-1 \in T_3(T) \text{ and } |T_3(T-1)| \leq C_1 - 1 | Y\}$$

$$= \sum_{r=C_1}^{T-1} C_1 \delta_0 \mathbb{P}\{|T_3(T)| = r + 1 | Y\}$$

$$= \sum_{r=C_1+1}^{T} C_1 \delta_0 \mathbb{P}\{|T_3(T)| = r | Y\}$$

$$= \sum_{r=0}^{T} \min\{r, C_1\} \delta_0 \mathbb{P}\{|T_3(T)| = r | Y\}, \tag{28}$$

where we add extra zeros for $r = 0, \ldots, C_1$. For event $Z$, we get

$$\sum_{r=0}^{T-1} \min\{r, C_1\} \delta_0 \mathbb{P}\{|T_3(T-1)| = r|Z\} + \delta_0 \mathbb{P}\{T - 1 \in T_3(T) \text{ and } |T_3(T-1)| \le C_1 - 1|Z\}$$

$$= \sum_{r=0}^{T-1} \min\{r, C_1\} \delta_0 \mathbb{P}\{|T_3(T-1)| = r|Z\}$$

$$= \sum_{r=0}^{T-1} \min\{r, C_1\} \delta_0 \mathbb{P}\{|T_3(T)| = r|Z\}$$

$$= \sum_{r=0}^{T} \min\{r, C_1\} \delta_0 \mathbb{P}\{|T_3(T)| = r|Z\}, \tag{29}$$

where in the first equation we use the notation of $Z$, in the second one we use $\mathbb{P}\{A|B\} = \mathbb{P}\{A \cap B|B\}$, and in the last equation we add $\mathbb{P}\{|T_3(T)| = T|Z\}$, which is equal to $0$. Multiplying (27), (28) and (29) by $\mathbb{P}\{X\}, \mathbb{P}\{Y\}, \mathbb{P}\{Z\}$, respectively, and summing up, we derive

$$\sum_{r=0}^{T-1} \min\{r, C_1\} \delta_0 \mathbb{P}\{|T_3(T-1)| = r\} + \delta_0 \mathbb{P}\{T - 1 \in T_3(T) \text{ and } |T_3(T-1)| \le C_1 - 1\}$$

$$= \sum_{r=0}^{T} \min\{r, C_1\} \delta_0 \mathbb{P}\{|T_3(T)| = r\}.$$

As a result, we have

$$\mathbb{P}\{E_T\} \ge 1 - \frac{T\delta}{K} - \sum_{r=0}^{T} \min\{r, C_1\} \delta_0 \mathbb{P}\{|T_3(T)| = r\}.$$

This concludes the inductive proof. In particular, taking $\delta_0 := \frac{\delta}{C_1}$ and

$$K \ge \frac{128^\alpha \ln\left(\frac{4K}{\delta}\right) (L_1 R_0)^\alpha}{9\delta_0} = C_1 \cdot \frac{128^\alpha \ln\left(\frac{4K}{\delta}\right) (L_1 R_0)^\alpha}{9\delta} = 64 \cdot 128^\alpha \cdot 160 \frac{(L_1 R_0)^{2+\alpha} \ln^2\left(\frac{4K}{\delta}\right)}{\delta},$$

we get that

$$\mathbb{P}\{E_K\} \ge 1 - \delta - \sum_{r=0}^{T} \min\{r, C_1\} \delta_0 \mathbb{P}\{|T_3(K)| = r\} \ge 1 - \delta - C_1 \delta_0 = 1 - 2\delta.$$

In particular, $E_K$ implies

$$R_K^2 \le 2R_0^2 - \frac{\gamma \lambda |T_3(K)|}{32L_1}.$$

What is more, we have

$$\sum_{k \in T_1(K) \cup T_2(K)} \gamma(f(x_k) - f^*) \le 2R_0^2 - \frac{\gamma \lambda |T_3(K)|}{32L_1}.$$

Therefore, we get

$$\frac{1}{K - |T_3(K)|} \sum_{k \in T_1(K) \cup T_2(K)} \gamma(f(x_k) - f^*) \le \frac{2R_0^2}{K - |T_3(K)|} - \frac{\gamma \lambda |T_3(K)|}{32L_1(K - |T_3(K)|)}.$$

Considering the RHS, it can be shown that it is the decreasing function of $|T_3(K)|$. Indeed, denoting

$$\phi(x) = \frac{2R_0^2}{K - x} - \frac{\gamma \lambda x}{32L_1(K - x)},$$

one can obtain

$$\phi'(x) = \frac{2R_0^2}{(K-x)^2} - \frac{\gamma\lambda K}{32L_1(K-x)^2} \leq 0$$

due to the lower bound on $K$. Consequently, we get

$$\frac{1}{K - |T_3(K)|} \sum_{k \in T_1(K) \cup T_2(K)} \gamma(f(x_k) - f^*) \leq \frac{2R_0^2}{K}.$$

Dividing both sides by $\gamma$, substituting (13), and lower bounding the LHS, we obtain that

$$\min_{k=0,\ldots,K-1} (f(x_k) - f^*) = \tilde{\mathcal{O}}\left(\max\left\{\frac{L_0 R_0^2}{K}, \frac{L_1 R_0^2 \sigma}{K^{\frac{\alpha-1}{\alpha}}}\right\}\right) \tag{30}$$

with $K = \Omega\left(\frac{(L_1 R_0)^{2+\alpha} \ln^2\left(\frac{4K}{\delta}\right)}{\delta}\right)$ holds with probability at least $1 - 2\delta$. Combining (26) and (30), we finish the proof. $\qquad\square$

## C    FROM THE BEST ITERATE TO A SINGLE IMPLEMENTABLE ITERATE

In Theorem 1 we state a bound for the best iterate over the "good" indices $J_K \coloneqq T_1(K) \cup T_2(K)$. This appendix shows that one can select a *single, fully implementable iterate* with (essentially) the same rate by sampling only $m = \lceil \log(1/\delta) \rceil$ uniformly random candidates from the run and picking the one with the smallest robust mini-batch estimate of $f$.

**Selection rule.**    Run CLIP-SGD for $K$ steps with the same $(\lambda, \gamma)$ as in Theorem 1. After that, do:

1. Sample $m = \lceil \log(1/\delta) \rceil$ indices independently and uniformly: $\tau_1, \ldots, \tau_m \sim \mathrm{Unif}\{0, 1, \ldots, K-1\}$.

2. For each $i \in \{1, \ldots, m\}$, form a robust mini-batch estimate $\hat{f}_i$ of $f(x_{\tau_i})$ using $B$ fresh i.i.d. samples $\{\xi_{i,b}\}_{b=1}^{B}$, independent of the training randomness, and any robust mean estimator valid under finite $\alpha$-moment noise (e.g., median-of-means or Catoni (Lugosi and Mendelson, 2019)).

3. Output $\hat{\tau} \in \arg\min_{i=1,\ldots,m} \hat{f}_i$ and return $x_{\hat{\tau}}$.

In order to show that the robust mean properly approximates the true function value, we introduce an additional assumption (needed only for this section).

**Assumption 4** (Loss $\alpha$-moment). *Let* $\alpha \in (1, 2]$ *be the same as in Assumption 3. There exists* $\sigma_f > 0$ *such that for all* $x \in \mathbb{R}^d$,

$$\mathbb{E}_\xi \big[ \, |f(x, \xi) - f(x)|^\alpha \, \big] \ \le \ \sigma_f^\alpha,$$

*where the samples* $\{\xi_{i,b}\}$ *used to compute* $\hat{f}_i$ *are drawn i.i.d. from the same distribution as in* (1) *and independently of the training process.*

**Theorem 3** (Single implementable iterate via randomization and robust evaluation). *Suppose Assumptions 1-4 hold. Run CLIP-SGD for* $K$ *steps with the* $(\lambda, \gamma)$ *of Theorem 1. Let* $m = \lceil \log(1/\delta) \rceil$. *There exists a constant* $C_\alpha > 0$ *(depending only on* $\alpha$ *and the chosen robust mean) such that if*

$$B \ \ge \ C_\alpha \Big( \frac{\sigma_f}{A} \Big)^{\frac{\alpha}{\alpha-1}} \log \frac{m}{\delta}, \qquad where \qquad A \coloneqq \frac{2R_0^2}{\gamma K},$$

*then with probability at least* $1 - 3\delta - \dfrac{C_1 \log(1/\delta)}{K}$ *we have*

$$f(x_{\hat{\tau}}) - f^\star \ \le \ 3A.$$

*In particular, substituting the choices of* $(\lambda, \gamma)$ *from Theorem 1 yields*

$$f(x_{\hat{\tau}}) - f^\star \ \le \ \tilde{\mathcal{O}} \bigg( \max \bigg\{ \frac{L_0 R_0^2}{K}, \ \frac{L_1 R_0^2 \sigma}{K^{(\alpha-1)/\alpha}} \bigg\} \bigg).$$

*Moreover, recalling* $C_1 = \Theta\big((L_1 R_0)^2 \log(4K/\delta)\big)$ *from Appendix B and the lower bound on* $K$ *in the large-radius regime, the success probability is at least* $1 - 4\delta$.

*Proof sketch.* Let $J_K \coloneqq T_1(K) \cup T_2(K)$. Appendix B shows that on the "good" event $E_K$ (which holds with probability at least $1 - 2\delta$),

$$\frac{1}{K - |T_3(K)|} \sum_{k \in J_K} \gamma \, (f(x_k) - f^\star) \ \le \ \frac{2R_0^2}{K}, \tag{31}$$

and $|T_3(K)| \le C_1$. Conditioning on $E_K$, draw $\tau_1, \ldots, \tau_m$ uniformly. With probability at least $1 - \frac{mC_1}{K}$ all $\tau_i \in J_K$. Equation (31) implies $\mathbb{E}[f(x_{\tau_i}) - f^\star] \le A \coloneqq 2R_0^2/(\gamma K)$ for any $\tau_i \in J_K$, and by Markov, $\Pr\big( \min_i f(x_{\tau_i}) - f^\star \le 2A \mid E_K \big) \ge 1 - \delta$ for $m = \lceil \log(1/\delta) \rceil$. Under Assumption 4, robust mean estimation over $B$ fresh samples gives $\max_i |\hat{f}_i - f(x_{\tau_i})| \le \varepsilon_B$ with probability at least $1 - \delta$, where $\varepsilon_B \le C_\alpha \sigma_f \big( \frac{\log(m/\delta)}{B} \big)^{1-1/\alpha}$. Choosing $B$ as in the statement ensures $\varepsilon_B \le A$, hence $f(x_{\hat{\tau}}) \le \min_i f(x_{\tau_i}) + A \le 3A$. A union bound completes the proof. $\square$

**Remarks.** (i) The validation does not modify optimization process; it adds only $mB = \tilde{\mathcal{O}}\big((\sigma_f/A)^{\frac{\alpha}{\alpha-1}} \log^2 \frac{1}{\delta}\big)$ extra stochastic loss evaluations. For $\alpha = 2$, this reduces to the familiar $B = \Theta\big((\sigma_f/A)^2 \log(m/\delta)\big)$. (ii) Assumption 4 is the loss-level analogue of Assumption 3 and is used only in this section to justify robust estimation of $f(x)$; all main theorems and proofs in the body remain unchanged. (iii) When $L_1 = 0$ or $K$ is much larger than the lower bound of Theorem 1 (large-radius case), the success probability simplifies to at least $1 - 4\delta$.

## D  NUMERICAL EXPERIMENTS

In this section, we present the results of numerical experiments on a toy example. As the convex function satisfying Assumption 2, we chose the one-dimensional function $f(x) = \|x\|_2^4$, for which $L_0$ and $L_1$ are known: $L_0 = 4$ and $L_1 = 3$, respectively. In order for the simulated stochastic gradient to satisfy Assumption 3, we, in turn, model the Clip-SGD iteration as follows:

$$x_{k+1} = x_k - \gamma \texttt{clip}(\nabla f(x_k) + s \cdot \xi_k, \lambda),$$

where

- $\gamma$ denotes the stepsize parameter;
- $\lambda$ denotes the clipping threshold;
- $\xi_k$ is a random variable generated from Lévy $\alpha$-stable distribution with parameters 1.1 and 0, which guarantees that Assumption 3 holds;
- $s$ is the scaling parameter of the stochasticity.

### D.1  WHICH TYPE OF CLIPPING PREVAILS DEPENDING ON THE LEVEL OF STOCHASTICITY?

The goal of the subsequent experiment is to *track* the more favorable clipping level depending on the stochasticity. That is, depending on whether parameter $s$ is large or small, we want to observe what clipping level is required for better convergence.

The baseline choice of the stepsize and clipping for $(L_0, L_1)$-smoothness in the deterministic case is given by the following values:

$$\gamma = 1/L_0 = 0.25; \qquad \lambda = L_0/L_1 = 1.33. \tag{32}$$

For the stochastic case, we selected empirical parameters similar to the dependence given in Theorem 1, without taking the scaling factor into account:

$$\gamma = 0.0003; \qquad \lambda = 80. \tag{33}$$

**Remark 3.** *Regarding* (33)*, the stepsize parameter $\gamma$ was chosen to be approximately theoretical; however, the clipping level was not – we considered several clipping levels, including the theoretical one, and selected the best. This does not contradict the theory, since the theoretical threshold $\lambda$ from Theorem 1 is chosen for an entire class of problems rather than for a specific problem.*

To demonstrate the trade-off between the deterministic and stochastic options for parameter selection, we consider two cases: $s = 0.01$ is the case with a low stochastic effect, and $s = 1000$ is the case with a high level of randomness. Results are provided below (see Fig. 1 and 2).

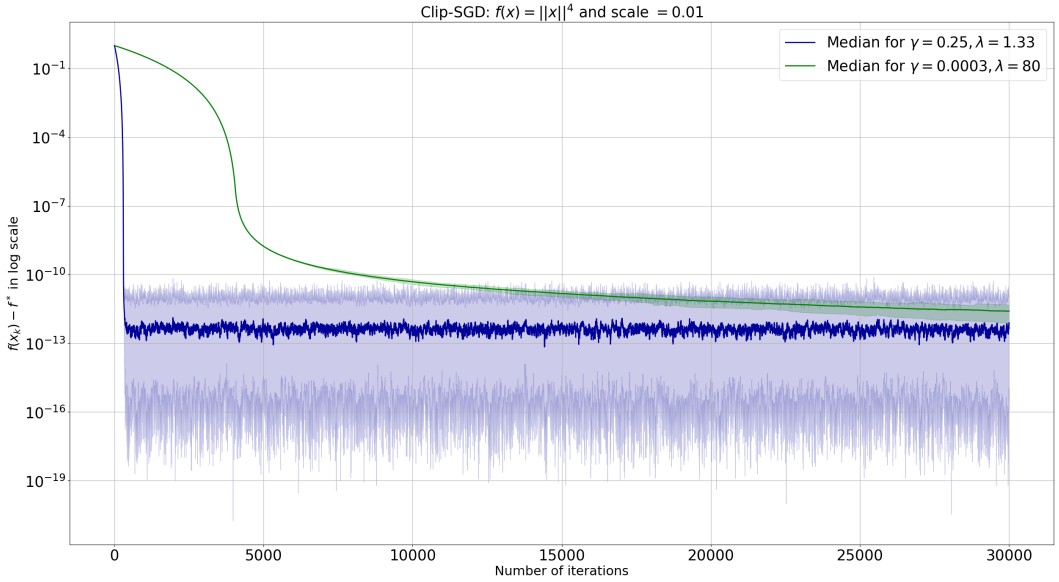

Figure 1: 100 runs of Clip-SGD with the low scale level

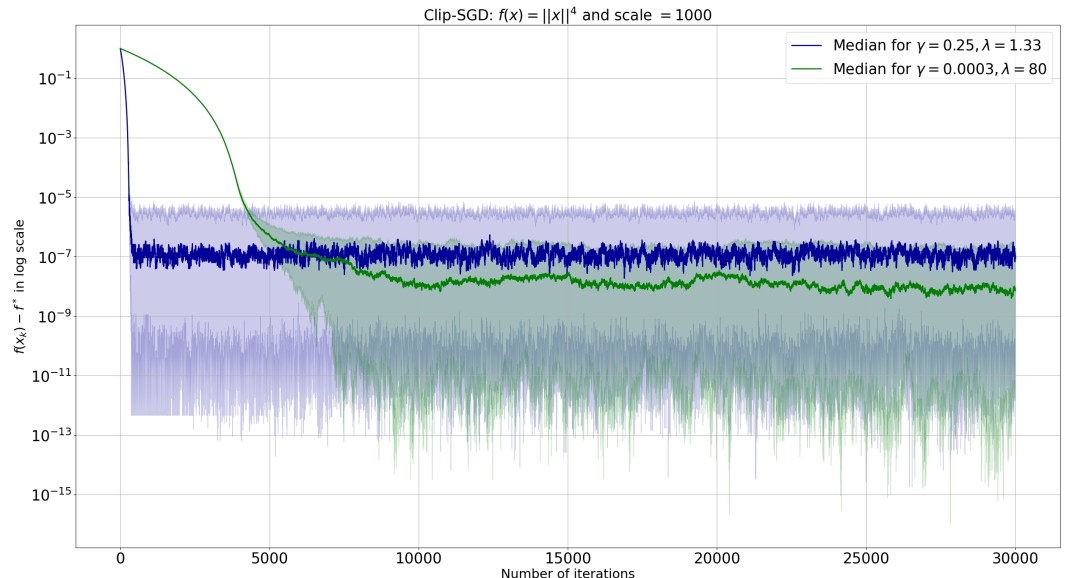

Figure 2: 100 runs of Clip-SGD with the high scale level

The experiment is designed as follows. We consider two parameter-selection cases: (32) corresponding to the deterministic setting, and (33) to the stochastic one. For each option, we run the Clip-SGD algorithm 100 times with 30k iterations, and the plots show medians of the runs (bold line), as well as the $5^{th}$ and $95^{th}$ percentiles (the boundaries of the shaded region of the corresponding color).

As we can observe, in the case of a small scaling factor $s$ (Fig. 1), the main issue is the generalized smoothness, since the effect of such stochasticity is negligible compared to the region of large gradients. At the same time, with a large scaling factor $s$ (Fig. 2), the stochasticity substantially complicates the problem, and its contribution to determining the length direction (since we consider one-dimentional problem) becomes stronger than the effect of the generalized smoothness.

As a result, this experiment confirms that there is a trade-off between the level of stochasticity and the region of large gradients. Thus, we have validated the importance of the problem addressed in our work and this is exactly demonstrated in Theorem 1.

### D.2 STANDARD CLIPPING VS INDEPENDENT SAMPLES FROM GAASH ET AL. (2025)

The next experiment compares the classical Clip-SGD with Clip-SGD using independent sampling for normalization and direction, as studied in Gaash et al. (2025). For clarity, we again examine different levels of stochastic scaling $s$ with the same choice of parameters (33). The design of the experiment remains the same as the previous one. Recall that in Gaash et al. (2025), the iteration is as shown in Algorithm 2. To model such behavior of the method, we consider the following iteration:

$$x_{k+1} = x_k - \gamma c_k(\nabla f(x_k) + s \cdot \xi_k); \qquad c_k = \min\left\{1, \frac{\lambda}{\|\nabla f(x_k) + s \cdot \eta_k\|}\right\},$$

where $\xi_k$ and $\eta_k$ are independent random variables from Lévy $\alpha$-stable distribution.

To compare these algorithms, we present convergence plots for $s = 0.01$ and $s = 1000$, respectively. As we can see, at a low level of stochasticity (Fig. 3), the algorithms behave almost identically, which is natural – in this case, the main role of clipping is to handle generalized smoothness; moreover, the stochastic gradient is very similar to the true one. Nevertheless, at a high level of stochasticity (Fig. 4), we observe that the independent sampling from Gaash et al. (2025) behaves very unstably due to the absence of noise normalization of $\xi_k$, which strongly affects the final magnitude of the direction.

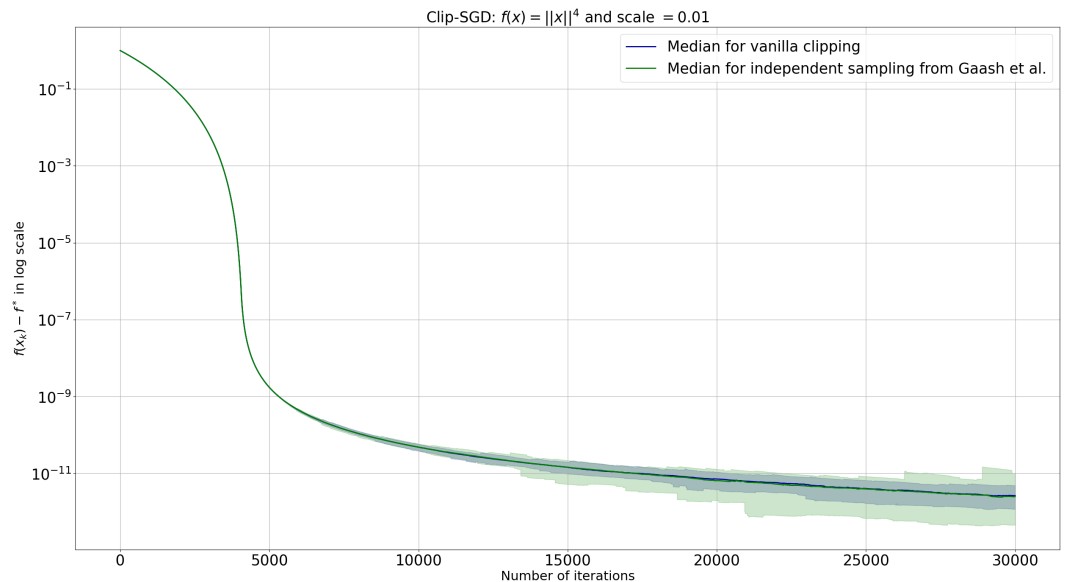

Figure 3: 100 runs of Clip-SGD and Clip-SGD with independent sampling for low scale level

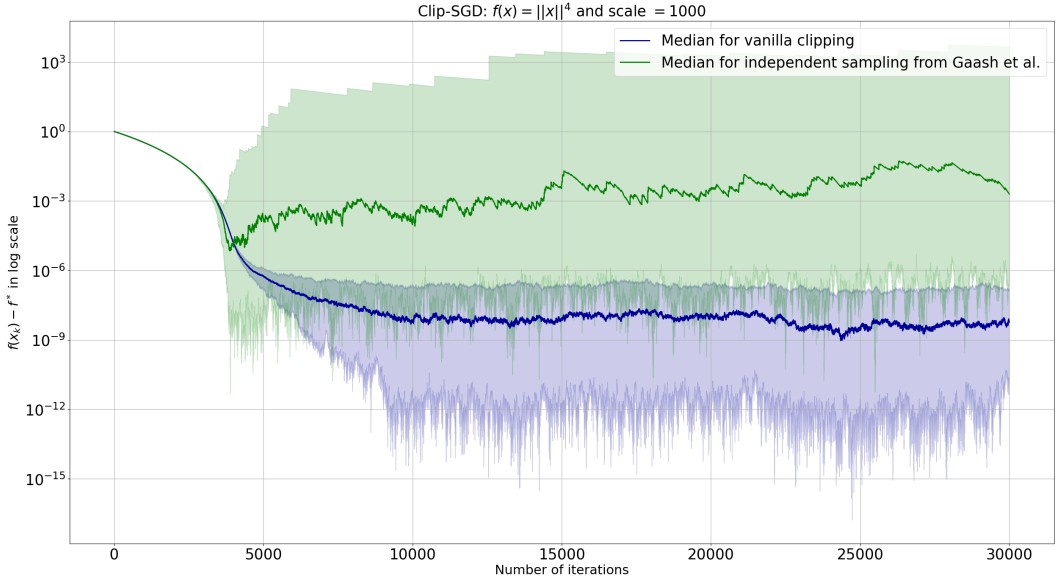

Figure 4: 100 runs of Clip-SGD and Clip-SGD with independent sampling for high scale level

This part of the experiments serves as direct confirmation of our reasoning in Section 5.1: clipping based on independent sampling can be a good option *only in the case of light-tailed stochasticity*, since the effect of noise on the true gradient becomes negligible. This means that the main role of clipping in Gaash et al. (2025) is to handle only $(L_0, L_1)$-smoothness. At the same time, the standard clipping can provide strong theoretical guarantees and practical significance, as this variant of gradient truncation is significantly more robust, even for toy examples.

