# OpenReview forum: "Convergence of Clipped-SGD for Convex $(L_0,L_1)$-Smooth Optimization with Heavy-Tailed Noise"
_ICLR.cc/2026/Conference — Submitted to ICLR 2026_

### Official Review · Reviewer_HYK1 · 2025-10-16

**Soundness:** 2
**Presentation:** 2
**Contribution:** 2
**Rating:** 4
**Confidence:** 3

**Summary:**

The authors study Clip‑SGD for convex objectives that satisfy $\left(L_0, L_1\right)$-smoothness under heavy‑tailed gradient noise. The authors aim to answer the question of how to choose the clipping level so that it simultaneously respects the geometry induced by $\left(L_0, L_1\right)$ and controls heavy tails (which typically demand growing thresholds).

**Strengths:**

1. High-probability convergence guarantees for Clip-SGD under both $\left(L_0, L_1\right)$-smoothness and heavy-tailed noise fills a gap, assuming that this is the first work to do so as is claimed.

2. The authors analyze standard Clip-SGD, which is both a strength and a weakness. As a strength, it is closer to what is done in practice, as a "first step" toward more complex analyses.

**Weaknesses:**

1. There is no empirical section, and the paper would possibly benefit greatly from even synthetic experiments that support their theory. There are no experiments to illustrate behavior versus unclipped SGD or double‑sample methods (even toy convex problems would be useful).

2. Could the authors discuss the primary technical obstacles in extending this high-probability analysis to the non-convex setting (that is, without assumption 1)?

**Questions:**

Please see weaknesses.

As a small suggestion, one work also may be worth looking into in the literature review is the paper [1], which also heavily studies clipping in the presence of heavy-tailed noise, though in a different setting than this paper.

[1] Lee et al., Efficient Distributed Optimization under Heavy-Tailed Noise. ICML, 2025.

---

> ### Author Response · Authors · 2025-11-18
> **Response to Reviewer HYK1**
>
> Dear Reviewer,
>
> Thank you for your time and feedback. We appreciate that you recognized that our results fill the gap in the literature and that we focus on the analysis of standard clipping, which is heavily used in practice. We are also open to further discussion regarding the shortcomings you pointed out or any questions you may still have.
>
> > ***W1:*** There is no empirical section, and the paper would possibly benefit greatly from even synthetic experiments that support their theory. There are no experiments to illustrate behavior versus unclipped SGD or double‑sample methods (even toy convex problems would be useful).
>
> ***A:*** Thank you for this remark. Generally speaking, we view our work as theoretical — we aimed to address questions regarding a mechanism that is already widely used in training deep learning models. Nevertheless, we agree that to improve clarity and to support our claims, it is worthwhile to provide numerical experiments, at least on a toy example. In the new version of the paper (see the updated PDF), in Appendix D, one can find the validated experiments for $f(x) = ||x||_2^4$ (with d = 1) in the following form:
>
> - The first study is devoted to examining the trade-off between clipping strategies for the deterministic and stochastic cases, depending on different scaling levels of the Lévy $\alpha$-stable distribution.
>
> - The second experiment serves as a confirmation of our conclusions from Section 5.1 regarding the comparison with the results from [10]. We show that independent sampling for normalization and direction behaves extremely unstably in the case of heavy stochasticity compared to standard clipping.
>
> For more detailed explanation of experiments, we refer to Appendix D.
>
> We would like to thank you for your suggestion -- it indeed allows us to empirically support the ideas presented in the paper.
>
> > ***W2:*** Could the authors discuss the primary technical obstacles in extending this high-probability analysis to the non-convex setting (that is, without assumption 1)
>
> ***A:*** Thank you for this comment. Speaking to the substance, the choice of the convex setting is motivated by two factors:
>
> 1. **The study of the convex case is relevant for modern research**. There are numerous recent works ([1], [2], [3] and [4]) showing that it is valid to consider convex analysis even from the perspective of training large models. Moreover, many popular techniques (one of the most notable examples being momentum [5], [6]) originated from the theoretical analysis of the convex setting and later migrated into algorithms applicable to the non-convex scenario.
>
> 2. **Generalized smoothness is not as well studied in the convex setting**. Despite the growing interest in $(L_0, L_1)$-smoothness, most works focus on the non-convex case. It has already been shown that, for example, normalization techniques such as Normalized SGD or SignSGD -- [7] (**Reviewer QWn5** clarified that we forgot to mention this work in our paper -- we added it in the updated version) and [8], respectively -- allow obtaining convergence guarantees either in expectation or with high probability under assumptions of heavy-tailed noise. It is also worth noting that proofs in the non-convex case usually lead to a deterministic term equal to $\mathcal{O}\Big(\frac{L_0\Delta_0}{poly(\varepsilon)}\Big)$, where $\Delta_0 = f(x_0) - f^*$ can be exponentially large -- this can slightly simplify the proof. For a more detailed discussion of exponential growth of $\Delta_0$, you can see the response to the **Reviewer QWn5**.
>
> For the reasons outlined above, we did not focus on the non-convex scenario -- we believe that the convex case is also important for the scientific community. Moreover, prior to our work, there were no studies on Clip-SGD method under the considered assumptions of generalized smoothness and a stochastic oracle. We believe that our work can serve as a foundation for further research, possibly extending to the non-convex case as well.
>
> Please refer to the following official comment for the notes on Q1.

---

> ### Author Response · Authors · 2025-11-18
> **Continuation of the response to Reviewer HYK1**
>
> This comment is a continuation of the previous one, and here we respond to Q1.
>
> > ***Q1:*** Please see weaknesses. As a small suggestion, one work also may be worth looking into in the literature review is the paper [9], which also heavily studies clipping in the presence of heavy-tailed noise, though in a different setting than this paper.
>
> ***A:*** Thank you for this clarification. We agree with the relevance of this work, as it provides theoretical guarantees for a distributed setup with a stochastic oracle under heavy-tailed noise. We will include this paper in the **Related Works** section.
>
> ---
>
> **References**
>
> [1] Kleinberg et al., An alternative view: When does SGD escape local minima? (ICML 2018)
>
> [2] Zhou et al., Sgd converges to global minimum in deep learning via star-convex path (ICLR 2019)
>
> [3] Liu et al., Loss landscapes and optimization in over-parameterized non-linear systems and neural networks (ACHA 2022)
>
> [4] Schaipp et al., The Surprising Agreement Between Convex Optimization Theory and Learning-Rate Scheduling for Large Model Training (ICML 2025)
>
> [5] Polyak, B. Some methods of speeding up the convergence of iteration methods (USSR Computational Mathematics and Mathematical Physics)
>
> [6] Nesterov, Y. A method for solving the convex programming problem with convergence rate O(1/k2) (Soviet Mathematics Doklady)
>
> [7] Liu Z., Zhou Z. Nonconvex stochastic optimization under heavy-tailed noises: Optimal convergence without gradient clipping (ICLR 2025)
>
> [8] Kornilov et al., Sign Operator for Coping with Heavy-Tailed Noise in Non-Convex Optimization: High Probability Bounds Under $(L_0, L_1)$-Smoothness (arXiv 2025)
>
> [9] Lee et al., Efficient Distributed Optimization under Heavy-Tailed Noise (ICML 2025)
>
> [10] Gaash et al., Convergence of Clipped SGD on Convex $(L_0,L_1)$-Smooth Functions (arXiv 2025)

---

> ### Comment · Reviewer_HYK1 · 2025-11-18
>
> Thank you for your detailed rebuttal. In particular, I appreciate the experiments that have been added in response as well as your clarifications on the positioning of the work. I believe they make the paper stronger and more substantiated. As most of my concerns are resolved, I have raised my score.

---

> > ### Author Response · Authors · 2025-11-19
> > **Response to Reviewer HYK1**
> >
> > Thank you very much for reviewing our responses and for raising your score. If you have any remaining questions or concerns, please let us know - we would be happy to provide further clarifications.

---

### Official Review · Reviewer_e9SL · 2025-10-25

**Soundness:** 3
**Presentation:** 3
**Contribution:** 2
**Rating:** 2
**Confidence:** 3

**Summary:**

The paper studies clipped stochastic gradient methods for convex $(L_0,L_1)$-smooth objectives under heavy-tailed noise with finite $\alpha$-th moments $(\alpha\in(1,2])$. It presents high-probability guarantees using a single-sample clipping scheme and claims improved comparisons to prior analyses (e.g., avoiding certain exponential dependences) together with brief illustrative experiments.

**Strengths:**

- The presentation connects $(L_0,L_1)$-smoothness with heavy-tailed noise in a single framework and recovers several special cases.

- Technically careful: the proofs are self-contained and the algorithmic template (standard clipping) is simple to implement.

- The organization is very clear.

**Weaknesses:**

- Problem novelty is weak. Both \emph{heavy-tailed} robustness for SGD (with clipping/truncation) and the \emph{$(L_0,L_1)$-smoothness} framework have been extensively studied; the paper largely resembles a \emph{combination} of two well-trodden threads (``A + B''), rather than introducing a new core idea or methodology.
- Topic saturation and maturity. Techniques used (clipping-based potential arguments, tail-sensitive concentration) are standard in this area; the contribution reads as a consolidation within known toolkits rather than a conceptual advance.
- Practical guidance is limited: the guarantees hinge on several constants and iteration thresholds (with explicit $\delta$-dependence), yet the paper does not delineate when its schedules outperform light-tailed baselines, nor provide actionable tuning rules when $\alpha$ and $L_1$ are unknown.

**Questions:**

While technically careful, the paper addresses a non-novel combination: $(L_0,L_1)$-smoothness and heavy-tailed robustness via clipping have each been widely covered, and the present work effectively composes these mature lines without a fresh idea that shifts the frontier.

Analyzing the advantages of Adam over SGD may be more interesting.

---

> ### Author Response · Authors · 2025-11-18
> **Response to Reviewer e9SL**
>
> Dear Reviewer,
>
> Thank you for your time and feedback. We appreciate that you recognized that our proofs are technically careful and the paper is clearly organized. We are also open to further discussion regarding the shortcomings you pointed out or any questions you may still have.
>
> > ***W1:*** Problem novelty is weak. Both heavy-tailed robustness for SGD (with clipping/truncation) and the $(L_0, L_1)$
> -smoothness framework have been extensively studied; the paper largely resembles a combination of two well-trodden threads ("A + B"), rather than introducing a new core idea or methodology.
>
>
>
> ***A:*** We kindly but firmly disagree with this point. It is not entirely clear what is meant by a “new core idea” or “methodology” in this context. Indeed, it would be misleading to describe our contribution as a mere combination of existing assumptions - in fact, such a combination has been considered **for the first time** in the context of the Clip-SGD method.
>
> While each of these assumptions has been individually studied in prior works, the key contribution of our paper lies precisely in highlighting the inconsistency in the choice of the clipping level when analyzing Clip-SGD. More specifically, under the assumption of generalized smoothness, the clipping level should remain constant (like $\Theta({L_0}/{L_1})$), whereas under heavy-tailed stochasticity, it must grow proportionally to the number of iterations. We explicitly pointed this out both before the contribution statement and in the **Discussion of the result** section, paragraph **Role of clipping**.
>
> Therefore, the essence of our work is not in reasserting the usefulness of clipping under these assumptions, but in providing **the first convergence analysis of Clip-SGD with high probability and without an exponential factor** $\exp(L_1R_0)$  under the most realistic and practically relevant set of assumptions via generalized smoothness and heavy-tailed noise.
>
> > ***W2:*** Topic saturation and maturity. Techniques used (clipping-based potential arguments, tail-sensitive concentration) are standard in this area; the contribution reads as a consolidation within known toolkits rather than a conceptual advance
>
> ***A:*** We kindly disagree with this statement. Indeed, the overall framework for analyzing Clip-SGD remains largely similar across works - it typically involves a descent lemma, bounding the bias and variance terms, and applying concentration inequalities. However, we explicitly emphasized the technical novelty in the **Main Result** section.
>
> The main difficulty in analyzing Clip-SGD in the convex setting lies in the fact that if one attempts to bound the gradient norm $||\nabla f(x_k)|| \leq \lambda/2$ through the inductive proof, this immediately reduces the analysis to the proofs covered by existing works on Clip-SGD (e.g., [1], [2]). Yet, such an approach inevitably leads to a bound containing an **exponential factor** $\exp(L_1R_0)$, which is prohibitive.
>
> To overcome this, we developed our analysis based on **stochastic sets** $T_1, T_2$ and $T_3$, which allow us to handle different regimes of the true gradient norm. This, in turn, implies that even though the bias and variance terms can be bounded, the **inductive step must be carried out with extreme care** - a logical construction that, to the best of our knowledge, **has not appeared** in any prior work.
>
> Therefore, we cannot agree that our proof relies merely on existing techniques.
>
> Please refer to the following official comment for the notes on W3 and Q1.

---

> > ### Author Response · Authors · 2025-11-18
> > **Continuation of the response to Reviewer e9SL**
> >
> > This comment is a continuation of the previous one, and here we respond to W3 and Q1.
> >
> > >***W3:*** Practical guidance is limited: the guarantees hinge on several constants and iteration thresholds (with explicit
> > $\delta$-dependence), yet the paper does not delineate when its schedules outperform light-tailed baselines, nor provide actionable tuning rules when $\alpha$ and $L_1$ are unknown.
> >
> > ***A:*** We appreciate the reviewer for pointing out this issue. Indeed, there is a well-known gap between theoretical guarantees and practical applicability - convergence analyses are typically derived under the assumption that certain problem parameters (such as $L_0, L_1, \alpha$) are known in advance. In practice, however, these parameters are unavailable and must be tuned empirically.
> >
> > Our work does not aim to provide a framework for adaptive parameter selection (although such analyses exist under standard smoothness assumptions). Instead, we focus on answering why Clip-SGD works in practice from a theoretical standpoint. In essence, we show that if a problem satisfies the classical assumptions we consider, then one can choose learning parameters such that the corresponding machine or deep learning model is guaranteed to converge.
> >
> > From a practical perspective, it has long been observed that Clip-SGD performs robustly in real-world applications. Moreover, similar behavior has been demonstrated for related methods such as Clip-Adagrad and Clip-Adam ([3], [4]), where introducing clipping into adaptive algorithms was shown to significantly improve training robustness.
> >
> > Regarding light-tailed baselines, we believe it is not entirely appropriate to compare different problem settings solely from the perspective of schedulers. However, we note that in Gaash [5], convergence proofs also require knowledge of problem parameters — $L_0, L_1, \sigma, \delta$.
> >
> > >***Q1:*** While technically careful, the paper addresses a non-novel combination:
> > $(L_0, L_1)$-smoothness and heavy-tailed robustness via clipping have each been widely covered, and the present work effectively composes these mature lines without a fresh idea that shifts the frontier. Analyzing the advantages of Adam over SGD may be more interesting.
> >
> > ***A:*** As noted above, this statement is not entirely accurate — the particular setup considered here is, to the best of our knowledge, being studied for Clip-SGD for the first time. We would understand the reviewer’s concern if our final convergence bound simply reproduced the classical smoothness results, but the bound we obtain is different. For that reason, we cannot agree with this claim.
> >
> > Regarding Adam, we agree that it could indeed be interesting to investigate. However, the results would most likely remain similar — to the best of our knowledge, the only work that studies Adam with clipping under heavy-tailed stochasticity and with high-probability guarantees is [4], where $L$-smoothness is assumed and the resulting convergence bounds fully reproduce those known for standard SGD. Moreover, [4] also demonstrates that clipping is essential in adaptive optimization schemes, further supporting this line of reasoning.
> >
> > ---
> >
> > **References**
> >
> > [1] Sadiev et al., High-Probability Bounds for Stochastic Optimization and Variational Inequalities: the Case of Unbounded Variance
> >
> > [2] Nguyen et al., Improved Convergence in High Probability of Clipped Gradient Methods with Heavy Tails (NeurIPS 2023)
> >
> > [3] Li, S., & Liu, Y., High probability analysis for non-convex stochastic optimization with clipping (arXiv 2023)
> >
> > [4] Chezhegov et al., Clipping Improves Adam-Norm and AdaGrad-Norm when the Noise Is Heavy-Tailed (ICML 2025)
> >
> > [5] Gaash et al., Convergence of Clipped SGD on Convex $(L_0,L_1)$-Smooth Functions (arXiv 2025)

---

> > > ### Author Response · Authors · 2025-11-27
> > >
> > > Dear Reviewer e9SL,
> > >
> > > We would like to respectfully follow up regarding our rebuttal. We understand the significant workload during the review period and appreciate the time you devote to evaluating submissions.
> > >
> > > If you have an opportunity, we would be grateful for any further comments or clarifications. Your feedback is highly valued.
> > >
> > > Thank you for your time and consideration.
> > >
> > > Best regards,
> > >
> > > Authors

---

### Official Review · Reviewer_aGea · 2025-10-30

**Soundness:** 3
**Presentation:** 4
**Contribution:** 3
**Rating:** 8
**Confidence:** 3

**Summary:**

This paper investigates the convergence of Clipped-SGD for convex optimization under $(L_{0},L_{1})$-smoothness and heavy-tailed noise where the gradient estimate has a bounded $\alpha$-th central moment for $\alpha \in (1, 2]$.

**Strengths:**

The authors identify a conflict in prior work: to handle $(L_{0},L_{1})$-smoothness, the clipping threshold $\lambda$ is typically set to a fixed constant, whereas to handle heavy-tailed noise, $\lambda$ needs to grow with the number of iterations $K$.
The main contribution of this paper is to bridge this gap, providing a high-probability convergence bound for Clipped-SGD under both conditions simultaneously with an unified clipping threshold strategy. This result also successfully avoids the exponential dependence on $L_{1}R_{0}$ that appeared in previous work.

The paper is in general well written, and the comparisons with related works are clear to me, especially the one with (Gaash et al. 2025), which make the technical contribution of the paper more clearer and interesting.

**Weaknesses:**

- Dependence on  $1/\delta$: To establish the high-probability bound, Theorem 1 (case 2) requires the total number of iterations $K = \Omega(\frac{(L_{1}R_{0})^{2+\alpha}}{\delta})$. This polynomial dependence on $1/\delta$ is not standard comparing with  $\log(1/\delta)$ in Theorem 1 (case 1).  Could the authors comment on whether it might be possible to use more advanced probabilistic tools to improve the dependency to $\log(1/\delta)$?

- As noted by the authors in the final section, the paper starts with the convex case, which is reasonable, and it would be also interesting to consider the non-convex case.

**Questions:**

- Could the authors comment on whether it might be possible to use more advanced probabilistic tools to improve the dependency to $\log(1/\delta)$ for Theorem 1 (case 2)?

- Does the analysis could be also applied to the more general noise models where $\sigma^2 = A(f(x) - f^*) + B \|\nabla f(x)\|^2 + C$ (Yu et al. 2025)?

- As the authors mentioned a lot on ``while in the presence of heavy-tailed noise, the threshold is often required to grow with the total number of iterations to ensure stability and convergence,'' it would be also good to add the threshold parameter in Table 1.

- Missing a “max” in the convergence rate of Thm 1(case 1)?

---

> ### Author Response · Authors · 2025-11-18
> **Response to Reviewer aGea**
>
> Dear Reviewer,
>
> Thank you for your time, feedback, and positive evaluation. We appreciate that you recognized the main contribution of our paper and the quality of writing. We are also open to further discussion regarding the shortcomings you pointed out or any questions you may still have.
>
> > ***W1:*** Dependence on $1/\delta$: To establish the high-probability bound, Theorem 1 (case 2) requires the total number of iterations $K = \Omega\Big(\frac{(L_1R_0)^{2 + \alpha}}{\delta}\Big)$. This polynomial dependence on $1/\delta$ is not standard comparing with $\log(1/\delta)$ in Theorem 1 (case 1). Could the authors comment on whether it might be possible to use more advanced probabilistic tools to improve the dependency to $\log(1/\delta)$?
>
> ***A:*** Thank you for highlighting this important point. It is worth noting that the factor $1/\delta$ is not as critical in our analysis compared to typically considered $\log(1/\delta)$ in high-probability settings. This is because, in classical analyses, $\log(1/\delta)$ usually appears jointly with $\varepsilon$, whereas in our final bound, factor $1/\delta$ arises without $\varepsilon$. This means that our result is stronger than what could potentially be obtained via a Markov-type inequality from an expectation-based bound, if such a bound existed. We also emphasized this explicitly in Section 5.2 to avoid misleading the reader. In other words, in the community, a bound of type $\mathcal{O}\Big(\frac{\log(1/\delta)}{\varepsilon}\Big)$ is regarded as “good,” while a bound of type $\mathcal{O}\Big(\frac{1}{\varepsilon\delta}\Big)$ is considered “poor”. Our bound $\mathcal{O}\Big(\frac{\log(1/\delta)}{\varepsilon} + \frac{1}{\delta}\Big)$ lies between the two — it is little bit weaker than $\mathcal{O}\Big(\frac{\log(1/\delta)}{\varepsilon}\Big)$ but stronger than $\mathcal{O}\Big(\frac{1}{\varepsilon\delta}\Big)$, since the factor $1/\delta$ appears separately from $\varepsilon$.
>
> Next, speaking about improving the convergence rate — that is, eliminating the factor $1/\delta$ — we have certain concerns in this regard. Let us consider three types of existing bounds: the deterministic one, the light-tailed bound, and our bound. It then turns out that:
>
> - **Deterministic case**. In [1], it is shown that for the analogue of gradient clipping - smoothed gradient clipping - the requirement $K = \Omega\big((L_1R_0)^{2}\big)$ is necessary.
> - **Light-tailed noise**. In Gaash [2], it is demonstrated that condition $K = \Omega\Big((L_1R_0)^{2}\log(1/\delta)\Big)$ is required under the assumption of light-tailed stochasticity.
> - **Heavy-tailed noise**. In our bound, $K = \Omega\Big(\frac{(L_1R_0)^{2 + \alpha}}{\delta}\Big)$ should hold.
>
> Our concern is that this term may, in fact, be non-improvable (except possibly for the replacement of $2 + \alpha$ with $2$). It may arise from the very nature of the stochasticity itself: the deterministic case yields a standard factor, sub-Gaussian noise introduces an additional logarithmic term $\log(1/\delta)$, while heavy-tailed stochasticity may naturally lead to a factor of type $1/\delta$. Unfortunately, it would be unfair to claim that our bound is indeed tight - as, to date, no probabilistic lower bounds are known for the problem of interest. Moreover, our current analysis of the iterates from $T_3(K)$ relies on the application of (conditional) Markov's inequality finitely many times with high probability. Therefore, it is not easy to see whether our analysis can be improved to eliminate $1/\delta$ dependence in this term.
>
> Please refer to the following official comment for the notes on W2, Q1, Q2, Q3 and Q4.

---

> > ### Author Response · Authors · 2025-11-18
> > **Continuation of the response to Reviewer aGea**
> >
> > This comment is a continuation of the previous one, and here we respond to W3, Q1, Q2, Q3 and Q4.
> >
> > > ***W2:*** As noted by the authors in the final section, the paper starts with the convex case, which is reasonable, and it would be also interesting to consider the non-convex case.
> >
> >
> > ***A:*** Thank you for this clarification. Referring to the existing results on generalized smoothness, the main difficulty typically arises in the convex setting, since an exponential factor $\exp(L_1R_0)$ may appear if the analysis is not handled carefully. In contrast, such an artifact does not occur in the non-convex case (see our response to **Reviewer QWn5**). This is one of the reasons why we chose to focus on the convex analysis.
> >
> > For example, in [3], the analysis is carried out for the non-convex case; however, existing convergence or complexity bounds with explicit dependence on $\Delta := f(x_0) - f^*$ suffer from potentially exponential growth related to $R_0$. We provided a detailed explanation of this phenomenon in our response to **Reviewer QWn5**. Therefore, in order to study a less explored setup (as the convex case is considered less frequently than the non-convex one under generalized smoothness), we posed the following question: is it possible to establish a high-probability convergence guarantee for the convex case under heavy-tailed stochasticity, while avoiding the emergence of the exponential factor? And, as a result, our work provides a positive answer to this question.
> >
> > >***Q1:*** Could the authors comment on whether it might be possible to use more advanced probabilistic tools to improve the dependency to $\log(1/\delta)$ for Theorem 1 (case 2)?
> >
> > ***A:*** Please, see our response to W1.
> >
> > >***Q2:*** Does the analysis could be also applied to the more general noise models where $\sigma^2 = A(f(x) - f^*) + B||\nabla f(x)||^2 + C$ (Yu et al. 2025)?
> >
> > ***A:*** We suppose the answer is yes - there already exist high-probability analyses that consider assumptions on the affine noise. However, we chose not to adopt such an assumption, since, to the best of our knowledge, there was currently no existing convergence analysis under generalized smoothness and heavy-tailed stochasticity in the convex setting. Therefore, we decided to adopt the classical heavy-tailed noise assumption. We leave this extension for the future work.
> >
> > >***Q3:*** As the authors mentioned a lot on "while in the presence of heavy-tailed noise, the threshold is often required to grow with the total number of iterations to ensure stability and convergence,'' it would be also good to add the threshold parameter in Table 1.
> >
> > ***A:*** Thank you for this clarification. We include the gradient clipping parameters in Table 1 in the updated version of the paper. For convenience, we have highlighted this change in magenta.
> >
> > >***Q4:*** Missing a “max” in the convergence rate of Thm 1(case 1)?
> >
> > ***A:*** Thank you for catching this typo. Indeed, the max was missing in the $\mathcal{O}(\cdot)$ notation for Case 1; we correct this in the updated version of the paper. For convenience, we have highlighted this change in magenta.
> >
> > ---
> >
> > **References**
> >
> > [1] Gorbunov et al., Methods for Convex $(L_0,L_1)$-Smooth Optimization: Clipping, Acceleration, and Adaptivity (ICLR 2025)
> >
> > [2] Gaash et al., Convergence of Clipped SGD on Convex $(L_0,L_1)$-Smooth Functions (arXiv 2025)
> >
> > [3] Kornilov et al., Sign Operator for Coping with Heavy-Tailed Noise in Non-Convex Optimization: High Probability Bounds Under $(L_0, L_1)$-Smoothness (arXiv 2025)

---

### Official Review · Reviewer_QWn5 · 2025-11-11

**Soundness:** 4
**Presentation:** 4
**Contribution:** 3
**Rating:** 4
**Confidence:** 4

**Summary:**

This paper studies stochastic optimization for convex and $(L_0, L_1)$ smooth functions under heavy-tailed gradient noise. Clipped-SGD is analyzed and the first high-probability bound is shown for Clipped-SGD for this problem class.

**Strengths:**

1. The high-probability bound of Clipped-SGD is derived for the first time for the considered problem.
2. The presented bounds recover the current best result when $L_1  = 0$.

**Weaknesses:**

1. The problem class and algorithm are both motivated by some deep learning in particular attention models, but the theoretical results are only presented for convex functions. It would be great if non-convex case can be studied.
2. The considered problem class (for more general nonconvex functions) has been studied in [1] and optimal in-expectation rate using normalized SGD has been derived. It would be helpful if this work can be compared with.
3. No numerical experiments presented.


[1] Liu, Zijian, and Zhengyuan Zhou. "Nonconvex Stochastic Optimization under Heavy-Tailed Noises: Optimal Convergence without Gradient Clipping." The Thirteenth International Conference on Learning Representations.

**Questions:**

1. Given that existing works have studied high-probability convergence for clipped-SGD for convex smooth functions, it would be helpful if the challenges in dealing with additional $(L_0, L_1)$ smoothness can be highlighted.

---

> ### Author Response · Authors · 2025-11-18
> **Response to Reviewer QWn5**
>
> Dear Reviewer,
>
> Thank you for your time and feedback. We appreciate that you recognized the novelty of the derived result. We are also open to further discussion regarding the concerns you pointed out or any questions you may still have.
>
> > ***W1:*** The problem class and algorithm are both motivated by some deep learning in particular attention models, but the theoretical results are only presented for convex functions. It would be great if non-convex case can be studied.
>
> ***A:*** Thank you for this clarification. Indeed, one of the main motivations for our study was the consideration of generalized smoothness and heavy-tailed stochasticity, both of which naturally arise in the analysis of deep learning models. Therefore, investigating the non-convex case would seem a natural next step. However, there are some reasons why we chose to focus on the convex setting instead:
>
> - **Relevance of convex analysis**. Recent studies highlight the continued relevance of convex analysis in contemporary research. For instance, works [1], [2], and [3] show that deep neural networks often exhibit locally convex-like behavior in certain regions of the parameter space. This observation suggests that convergence analyses developed for convex functions can, to some extent, be applied to modern deep learning problems. Furthermore, studies such as [11] demonstrate that convex optimization theory aligns well with large-scale model training, particularly regarding learning-rate schedules.
>
> - **Lower degree of prior exploration**. Compared to the convex case, generalized smoothness appears to be significantly better studied in the non-convex setting, even under heavy-tailed noise. For example, one may point to work [4], which you mentioned, or to work [7], where a sign-based estimator yields high-probability convergence guarantees for heavy-tailed stochasticity in the non-convex case. Despite this, convex analysis has always played a central role in optimization methods: many techniques that originally emerged from the convex setting continue to be widely used both in theoretical proofs and in practical algorithms for non-convex problems. There are quite numerous examples - from momentum techniques [8], [9] to local-update methods in distributed optimization [10].
>
> - **Hidden exponentially large factors**. One of the main motivations of our work is to derive convergence guarantees that are *independent* of the exponentially large factors involving $L_1 R_0$, where $R_0 = \|\| x_0 - x^* \|\|$. However, in the non-convex setting, all existing convergence or complexity bounds (including those in [4]) that we are aware of explicitly depend on $\Delta := f(x_0) - f^* $, which can be exponentially larger than $R_0$ when $x^* $ exists.  For example, for $f(x) = \|\| x \|\|^4$ and $\|\| x_0 \|\| = 100$, we have $R_0 = 100$, while $\Delta = 10^8$, which is much larger than $R_0$. The situation becomes even worse for functions $f(x)$ that grow exponentially fast as $\|\|x - x^* \|\| \to \infty$. That is, a straightforward application of non-convex results to the convex case leads to highly suboptimal guarantees.
>
> Therefore, since the convex setting is both theoretically important and still practically relevant, we decided to focus our analysis on the convex scenario.
>
> > ***W2:*** The considered problem class (for more general nonconvex functions) has been studied in [4] and optimal in-expectation rate using normalized SGD has been derived. It would be helpful if this work can be compared with.
>
> ***A:*** Thank you for pointing out this work. It is indeed relevant to the topic under investigation. We have included it in the Related Works section. For convenience, we have highlighted the text preceding the added reference in magenta. However, we are unable to provide a direct comparison similar to that with Gaash et al. [5], since [4] presents an analysis for the non-convex case and in terms of the expectation. As mentioned in our previous response, the non-convex analysis under generalized smoothness is investigated better than the convex one and non-convex results implicitly contain exponentially large factors of $R_0$ when applied to the convex problems. Therefore, Gaash et al. remains the closest work and the most appropriate baseline for comparison in the high-probability convex setting.
>
> Please refer to the following official comment for the notes on W3 and Q1.

---

> > ### Author Response · Authors · 2025-11-18
> > **Continuation of the response to Reviewer QWn5**
> >
> > This comment is a continuation of the previous one, and here we respond to W3 and Q1.
> >
> > >***W3:*** No numerical experiments presented.
> >
> > ***A:*** We interpret our work as theoretical, since we aim to explain why Clip-SGD works under a more realistic setup that has not been previously considered for this algorithm. However, we agree that numerical experiments, even simple ones, can fully support the logic of our exposition. Following the suggestion of **Reviewer HYK1**, we added an experimental section to the appendix (see Appendix D), in which, for a toy example, we capture the trade-off between clipping levels depending on the level of stochasticity -- this directly reflects the essence of the clipping choice in Theorem 1. Moreover, we compared standard clipping with the clipping based on independent sampling from [5], thereby confirming our comparison from Section 5.1.
> >
> > It is worth noting that we do not see much value in adding numerical experiments involving the training of large models -- such experiments have been conducted in many works, and the superiority of clipped versions of the algorithms has long been well established.
> >
> > >***Q1:*** Given that existing works have studied high-probability convergence for clipped-SGD for convex smooth functions, it would be helpful if the challenges in dealing with additional $(L_0,L_1)$-smoothness can be highlighted.
> >
> > ***A:*** Thank you for this remark. The main difficulty in our analysis is controlling the number of iterations for which the true gradient is large; in our notation this corresponds to $||\nabla f(x_k)|| > \frac{\lambda}{2}$, which is equivalent to $k \in T_3(K)$. In a deterministic analysis one can show gradient decay for Smoothed Gradient Clipping [6], which identifies the iterations where the gradient norm is large. In the setting of [5], due to independent samples used for normalization and direction, one can write the concentration of the numerator and denominator separately, which makes the situation close to the deterministic one.
> >
> > However, when using the standard gradient clipping in the presence of heavy-tailed noise, one must carefully control the count of large gradients - this is captured by the set $T_3(K)$. Doing so substantially complicates the inductive step, and this inductive argument is, to the best of our knowledge, novel compared to existing techniques. We have also provided a detailed discussion of these aspects in the **Main Result** section of the initial version of the paper.
> >
> > ---
> >
> > **References**
> >
> > [1] Kleinberg et al., An alternative view: When does SGD escape local minima? (ICML 2018)
> >
> > [2] Zhou et al., Sgd converges to global minimum in deep learning via star-convex path (ICLR 2019)
> >
> > [3] Liu et al., Loss landscapes and optimization in over-parameterized non-linear systems and neural networks (ACHA 2022)
> >
> > [4] Liu Z., Zhou Z. Nonconvex stochastic optimization under heavy-tailed noises: Optimal convergence without gradient clipping (ICLR 2025)
> >
> > [5] Gaash et al., Convergence of Clipped SGD on Convex $(L_0,L_1)$-Smooth Functions (arXiv 2025)
> >
> > [6] Gorbunov et al., Methods for Convex $(L_0,L_1)$-Smooth Optimization: Clipping, Acceleration, and Adaptivity (ICLR 2025)
> >
> > [7] Kornilov et al., Sign Operator for Coping with Heavy-Tailed Noise in Non-Convex Optimization: High Probability Bounds Under $(L_0, L_1)$-Smoothness (arXiv 2025)
> >
> > [8] Polyak, B. Some methods of speeding up the convergence of iteration methods (USSR Computational Mathematics and Mathematical Physics)
> >
> > [9] Nesterov, Y. A method for solving the convex programming problem with convergence rate O(1/k2) (Soviet Mathematics Doklady)
> >
> > [10] Stich, S. Local SGD Converges Fast and Communicates Little (ICLR 2019)
> >
> > [11] Schaipp et al., The Surprising Agreement Between Convex Optimization Theory and Learning-Rate Scheduling for Large Model Training (ICML 2025)

---

> > > ### Author Response · Authors · 2025-11-27
> > >
> > > Dear Reviewer QWn5,
> > >
> > > We would like to respectfully follow up regarding our rebuttal. We understand the significant workload during the review period and appreciate the time you devote to evaluating submissions.
> > >
> > > If you have an opportunity, we would be grateful for any further comments or clarifications. Your feedback is highly valued.
> > >
> > > Thank you for your time and consideration.
> > >
> > > Best regards,
> > >
> > > Authors

---

### Meta-Review · Area_Chair_5UFr · 2026-01-08

**Summary:**

* Scope and Generalization: Several reviewers (QWn5, HYK1) noted that while the problem is motivated by deep learning (e.g., attention models), the theoretical analysis is restricted to convex functions.

* Novelty and Contribution: Reviewer e9SL argued that the paper is a simple combination of two existing topics (heavy-tailed noise and $(L_0, L_1)$-smoothness) without introducing a fresh conceptual advance.

* Empirical Validation: Multiple reviewers (QWn5, HYK1, e9SL) initially pointed out a total lack of numerical experiments to support the theoretical claims or compare the method against baselines.

* Technical Dependencies: Reviewer aGea highlighted a non-standard polynomial dependence on $1/\delta$ in the high-probability bound (Case 2), which is typically logarithmic ($log(1/\delta)$) in standard analyses.

* Practicality: Reviewer e9SL mentioned that the algorithm requires knowledge of unknown constants (like $\alpha$ and $L_1$), limiting its practical guidance.

**Reviewer Concerns:**

Addressed Concerns:

* Empirical Validation: The authors added a new section (Appendix D) with toy experiments showing the trade-off in clipping levels and comparing standard clipping to double-sampling methods. Reviewer HYK1 explicitly stated this resolved their concern.

* Comparison with Related Work: The authors incorporated and discussed several missing references to better position their work.

* Clarification of Convexity: The authors provided a technical justification for focusing on the convex case: namely, that non-convex analyses often hide exponential factors ($exp(L_1 R_0)$) that their convex analysis specifically aims to avoid.

Outstanding Concerns

* Non-convex Analysis: While the authors justified their focus, they did not actually provide a non-convex analysis, leaving this as future work.

* Tightness of $1/\delta$ Dependence: The authors admitted it is unclear if the $1/\delta$ factor can be improved to $log(1/\delta)$ because no probabilistic lower bounds currently exist for this specific setting.

* Novelty Perception: Reviewer e9SL’s fundamental critique that the work is a combination of known results remains a point of disagreement, as the reviewer did not engage further to acknowledge the technical novelties (stochastic sets $T_1, T_2, T_3$) claimed by the authors.

**Reviewer Scores:**

* HYK14: Positive. The reviewer explicitly increased their score after the authors added experiments and clarified the work's positioning.

* aGea: Though they did not reply to the rebuttal, their initial review was positive. The authors provided a detailed explanation for the $1/\delta$ factor, which likely would have satisfied this reviewer's curiosity.

* QWn5: The authors believe QWn5 would have increased their score if the system allowed it, as the reviewer's main technical concern (lack of experiments) was addressed via the same update that satisfied HYK1

* e9SL: This reviewer expressed "serious concerns" about novelty and relevance. While the authors corrected some misinterpretations, the fundamental disagreement on whether the work has substantial contribution suggests the score would remain negative.

---

### Decision · Program_Chairs · 2026-01-26

Reject